# On Accelerating Diffusion-based Molecular Conformation Generation in SE(3)-invariant Space

## Abstract

Diffusion-based generative models in SE(3)-invariant space have demonstrated promising performance in molecular conformation generation, but typically require solving stochastic differential equations (SDEs) with thousands of update steps. Till now, it remains unclear how to effectively accelerate this procedure explicitly in SE(3)-invariant space, which greatly hinders its wide application in the real world. In this paper, we systematically study the diffusion mechanism in SE(3)-invariant space via the lens of approximate errors induced by existing methods. Thereby, we develop more precise approximate in SE(3) in the context of projected differential equations. Theoretical analysis is further provided as well as empirical proof relating hyper-parameters with such errors. Altogether, we propose a novel acceleration scheme for generating molecular conformations in SE(3)-invariant space. Experimentally, our scheme can generate high-quality conformations with 50x–100x speedup compared to existing methods. Code is open-sourced at https://anonymous.4open.science/r/Fast-Sampling-41A6.

## 1 Introduction

Given an atom-bond molecular graph, conformation generation asks for sampling viable 3-dimensional coordinate configurations of atoms following the Boltzmann distribution of its compositional free energy (Strodel, 2021). As the 3-dimensional conformation generally determines a molecule's macroscopic properties, conformation generation is a basic yet essential task in cheminformatics, drug discovery, and material engineering. Traditional solutions rely on optimizing over potential energy surface (e.g., force field (Riniker & Landrum, 2015) and density functional theory (Castro et al., 2004)), which suffer from a variety of drawbacks separately, such as low coverage, high computational complexity, and heavy demand for prior knowledge.

Recently, there has been an increasing interest in applying diffusion-based generative models to sample molecular conformations (Shi et al., 2021; Zhang et al., 2023; Luo et al., 2021; Xu et al., 2022; Jing et al., 2022; Zhou et al., 2023; Fan et al., 2023a;b), as this line of work demonstrates strong modeling ability to capture the wide conformation distribution, analogous to their counterparts in computer vision area (Ho et al., 2020b; Dhariwal & Nichol, 2021). A central factor of such success lies in incorporating the roto-translational property of SE(3)-invariant space (De Bortoli et al., 2022), which is intrinsically equipped in 3-dimensional geometry. Unfortunately, unlike counterparts in computer vision, diffusion-based conformation generators that explicitly incorporate SE(3) property typically cost up to several thousand sampling steps. Till now it is vague how to directly accelerate them with standard solvers (e.g., DPM solver and high-order solvers (Song et al., 2020b; Dormand & Prince, 1980; Lu et al., 2022a)), while enforcing such solvers may result in unmeaningful conformations, and the reason causing this failure remains unclear. Although some other methods may be easily accelerated via bypassing this SE(3) riddle with much prior knowledge (Ganea et al., 2021; Jing et al., 2022), we believe that getting to the bottom of SE(3) can provide insightful perspectives to the future research.

In this paper, we systematically investigate how to effectively accelerate diffusion-based generation in SE(3)-invariant space for molecule generation tasks. To this end, we analyze current modeling methods in SE(3) (Shi et al., 2021; Xu et al., 2022; Zhou et al., 2023) and theoretically pose crucial

mistakes shared in these methods, which inevitably bring about the failure of acceleration. From the perspective of differential geometry and projected differential equations, we correct these mistakes and propose more accurate approximations of score functions with a provably tight bound. This approximation is designed to mimic a projection into SE(3)-invariant space. As such, we propose a plausible scheme to accelerate diffusion-based generation on top of the corrected modeling with a hyper-parameter. We further empirically demonstrate the relationship between hyper-parameter and the model's prediction error, which we believe can provide useful suggestions for future solver design. Extensive experiments are conducted on QM9 and Drugs datasets (Axelrod & Gomez-Bombarelli, 2022). Our acceleration scheme can sample high-quality molecular conformations by slightly modifying GeoDiff (Xu et al., 2022) and SDDiff (Zhou et al., 2023), but with 50–100x speedup. In summary, our contributions are:

- We analyze the modeling mistakes occurring in the current SE(3)-based methods and present effective approximation to correct them therein.
- We give a theoretically tight bound of our approximation, which also empirically performs well.
- We for the first time propose a plausible scheme for accelerating diffusion-based molecular conformation generation in SE(3)-invariant space, achieving remarkable speedup without sacrificing sampling quality.

## 2 RELATED WORKS

**Molecular conformation generation.** Various methods have been proposed to generate molecular conformers. Some notable approaches include GeoMol (Ganea et al., 2021), which utilizes Graph Neural Networks (GNNs) to predict local structures and connectivity information for subsequential manual assembly of conformers. In contrast, DMCG (Zhu et al., 2022) offers an end-to-end solution by directly predicting atom coordinates while maintaining roto-translational invariance through an SE(3)-invariant loss function. Recently, a growing interest has emerged in diffusion-based methodologies (see Appendix A). To simplify the analysis of SE(3)-invariance, some methods shift the modeling from atom coordinates to pairwise distances. A subset of them (Shi et al., 2021; Zhang et al., 2023; Luo et al., 2021) introduce perturbations to inter-atomic distances and subsequently estimate the corresponding coordinate scores. On the other hand, two most closely related works to the present study, GeoDiff (Xu et al., 2022) and SDDiff (Zhou et al., 2023) both choose to perturb atomic coordinates, but their distance distribution modelings are distinct. Other models focuse on molecular local structure designs. FrameDiff (Yim et al., 2023) parameterizes the diffusion of the frame translation and torsion angles by considering diffusion on SE(3)-manifold. Torsional diffusion (Jing et al., 2022) further eases the problem by applying RDkit to first generate local structures so that the lengths of atom bounds and then apply the diffusion process on torsion angles. Another different method, EC-Conf (Fan et al., 2023b), is a consistency model which can transform the conformation distribution into a noise distribution with a tractable trajectory satisfied SE(3)-equivariance.

**Sampling acceleration of diffusion-based models.** The reverse diffusion process typically takes thousands of steps. To accelerate the diffusion process, some new diffusion models such as consistency models (Song et al., 2023) are proposed. Consistency models directly map noise to data, enabling the generation of images in only a few steps. EC-Conf (Fan et al., 2023b) represents an application of consistency models in substantial reduced number of steps for molecular conformation generation. A more naive approach is simply reducing the number of sampling iterations. DDIM (Song et al., 2020a) uses a hyper-parameter to control the sampling stochastical level and finds that the decrease of the reverse iteration number results improved sample quality due to less stochasticity. This phenomenon can be attributed to the existence of a probability ODE flow associated with the stochastic Markov chain of the reverse diffusion process (Song et al., 2020b). This implies that several numerical ODE solver methods can be applied to solve the reverse diffusion. DPM-solver (Lu et al., 2022a) leverages the semi-linear structure of probability flow ODE to develop customized ODE solvers, and also provides high-order solvers for the probability ODE flow which can generate high-quality samples in only 10 to 20 iterations. Then DPM is extended as DPM-solver++ (Lu et al., 2022b) for sampling with classifier-free guidance. However, such a dedicated solver can only be applied in Euclidean space. To our best knowledge, there is no diffusion solver for SE(3)-invariant space or on the pairwise distance manifold. Hence, accelerating SE(3)-diffusion process remains a challenge.

## 3 PRELIMINARY

### 3.1 MOLECULAR CONFORMATION GENERATION

Given a specific molecular graph $G$, the molecular conformation generation task aims to generate a series of independent and identically distributed sample conformations $\mathcal{C}$ from the conditional probability distribution denoted as $p(\mathcal{C}|G)$. In this context, the distribution function $p$ adheres to the underlying Boltzmann distribution (Noé et al., 2019).

Each molecule is represented as an undirected graph, denoted as $G = (V, E)$, where the set $V$ represents the ensemble of atoms within the molecule, while $E$ signifies the collection of inter-atomic chemical bonds. Additionally, the graph incorporates node features $\boldsymbol{h}_v \in \mathbb{R}^f$ for all nodes $v \in V$ and edge features $\boldsymbol{e}_{uv} \in \mathbb{R}^{f'}$ for all edge connections $(u, v) \in E$. These features encapsulate information about atom types, formal charges, and bond types, among other characteristics.

To streamline the notation, the set of atoms in three-dimensional Euclidean space is represented as $\mathcal{C} = [\mathbf{x}_1, \dots, \mathbf{x}_n] \in \mathbb{R}^{n \times 3}$, and the distance between nodes $u$ and $v$ is expressed as $d_{uv} = \|\mathbf{x}_u - \mathbf{x}_v\|$. To model the generative process effectively, the generative model is denoted as $p_{\boldsymbol{\theta}}(\mathcal{C}|G)$.

### 3.2 EQUIVARIANCE WITHIN MOLECULAR CONFORMATION ANALYSIS

Equivariance with respect to translation and rotation operations, defined by the SE(3) groups, holds significant interdisciplinary relevance across various physical systems. Therefore, it assumes a pivotal role in the modeling and analysis of three-dimensional geometric structures, as highlighted in prior research (Thomas et al., 2018; Weiler et al., 2018; Chmiela et al., 2019; Fuchs et al., 2020; Miller et al., 2020; Simm et al., 2020; Batzner et al., 2022). In mathematical terms, a model $\mathbf{s}_{\boldsymbol{\theta}}$ is considered equivariant concerning the SE(3) group if it satisfies the condition $\mathbf{s}_{\boldsymbol{\theta}}(T_f(\mathbf{x})) = T_g(\mathbf{s}_{\boldsymbol{\theta}}(\mathbf{x}))$ for any arbitrary transformations $f$ and $g$ belonging to the SE(3) group. An effective strategy is to employ inter-atomic distances, which naturally exhibit equivariance with respect to the SE(3) groups (Shi et al., 2021; Xu et al., 2022; Gasteiger et al., 2020).

### 3.3 PAIRWISE-DISTANCE MANIFOLD

Pairwise-distance matrices (adjacent matrices) lie in a sub-manifold of $\mathbb{R}_+^{n \times n}$. A pairwise-distance matrix $d = [d_{ij}] \in \mathbb{R}_+^{n \times n}$ is said to be valid if there is a set of coordinates $\mathcal{C} = [\mathbf{x}_1, \dots, \mathbf{x}_n]$ s.t. $d_{ij} = \|\mathbf{x}_i - \mathbf{x}_j\|, \forall i, j = 1, \dots, n$. The manifold of valid distance matrices is a proper sub-manifold of $\mathbb{R}_+^{n \times n}$. Directly applying the diffusion process, i.e., $\tilde{d} = d + \mathbf{z}$ for some $\mathbf{z} \sim \mathcal{N}(\mathbf{0}_{n \times n}, \boldsymbol{I})$ would result in an invalid pairwise-distance $\tilde{d}$. Meanwhile, in the reverse process of diffusion, enforcing the model to generate a feasible distance matrix is non-trivial. Some previous works (Hoffmann & Noé, 2019) utilize the spectral theorem to generate valid pairwise distance but such a method involves matrices decomposition, which would cause huge computational cost. In some other works, authors implicitly assume that the manifold of pairwise distance is surjective to $\mathbb{R}_+^{n \times n}$, resulting in inaccurate computation of the score function (Shi et al., 2021; Xu et al., 2022; Zhou et al., 2023).

## 4 METHOD

In this work, we aim to explicitly accelerate the sampling of SE(3)-invariant diffusion models. Surprisingly, our scheme only requires **two slight modifications** on top of existing models, i.e., GeoDiff (Xu et al., 2022) and SDDiff (Zhou et al., 2023), to enable efficient sampling in much fewer steps: 1) replacing "summation" with "mean" term in estimating the score (Eq. 5) and 2) multiplying a factor "scale" to correlate the error shift (Eq. 14).

We consider SE(3)-invariant diffusion models that model inter-atomic distances. Specifically, given a molecular conformation $\mathcal{C}_0 \in \mathbb{R}^{n \times 3}/\text{SE}(3)$ and let $\mathcal{C}_0$ be embedded by a $n \times 3$ matrix, i.e., $\mathcal{C}_0 = [\mathbf{x}_1, \dots, \mathbf{x}_n] \in \mathbb{R}^{n \times 3}$, they define a forward diffusion process of the conformation embedding (we consider $\mathcal{C}_0$ as a vector to simplify our notations) (Song et al., 2020b)

$$q_{0t}\left(\mathcal{C}_t \mid \mathcal{C}_0\right) = \mathcal{N}\left(\mathcal{C}_t \mid \mathcal{C}_0, \sigma_t^2 \boldsymbol{I}\right) \quad \Leftrightarrow \quad \partial \mathcal{C}_t = \sqrt{\frac{\mathrm{d}\sigma_t^2}{\mathrm{d}t}} \partial \mathbf{w}_t \tag{1}$$

where $\mathcal{C}_0 \sim q_0(\mathcal{C}_0)$ and $q_0$ is the distribution of the dataset, $\sigma_t = \sigma(t)$ is an increasing function, and the corresponding ODE reverse flow is

$$\frac{\partial \mathcal{C}_t}{\partial t} = -\frac{1}{2}\frac{\mathrm{d}\sigma_t^2}{\mathrm{d}t}\nabla_{\mathcal{C}_t} \log q_{0t}\left(\mathcal{C}_t \mid \mathcal{C}_0\right), \quad \mathcal{C}_T \sim q_{0T}\left(\mathcal{C}_T \mid \mathcal{C}_0\right) \approx q_T\left(\mathcal{C}_T\right), \tag{2}$$

To define an equivariant reverse process, we need to compute an equivariant score function $\nabla_{\mathcal{C}_t} \log q_{0t}(\mathcal{C}_t|\mathcal{C}_0)$. By assuming that the manifold dimension of the pairwise distance is $n^2$, existing methods use the chain rule to compute (Shi et al., 2021; Xu et al., 2022; Zhou et al., 2023)

$$\nabla_{\mathcal{C}_t} \log q_{0t}(\mathcal{C}_t|\mathcal{C}_0) \approx \sum_{i=1}^{n} \sum_{j \in N(i)} \frac{\partial d_{ij}^{(t)}}{\partial \mathcal{C}_t} \nabla_{d_{ij}^{(t)}} \log q_{0t}(d_t|d_0) \tag{3}$$

where $d = [d_{ij}] = [\|\mathbf{x}_i - \mathbf{x}_j\|]$. They train the model $\mathbf{s}_{\boldsymbol{\theta}} = \mathbf{s}_{\boldsymbol{\theta}}(\mathcal{C}_t, t, \mathcal{G}) = \mathbf{s}_{\boldsymbol{\theta}}(d_t, t, \mathcal{G})$ satisfying $\mathbf{s}_{\boldsymbol{\theta}}(d_t, t, \mathcal{G}) = -\sigma_t \nabla_{\mathcal{C}_t} \log q_{0t}(\mathcal{C}_t|\mathcal{C}_0)$ and $\mathcal{G}$ represents the molecular graph. In the following, for notation simplification, we omit the input $\mathcal{G}$ and $\mathbf{s}_{\boldsymbol{\theta}}(d_t, t) = \mathbf{s}_{\boldsymbol{\theta}}(d_t, t, \mathcal{G})$.

We find that applying the usual discretilization (Lu et al., 2022a) (see Appendix B)

$$\mathcal{C}_t \approx \mathcal{C}_s + [\sigma(t) - \sigma(s)]\,\mathbf{s}_{\boldsymbol{\theta}}(d_s, s) \tag{4}$$

cannot produce substantial conformations. To enable efficient sampling, we apply two core modifications. The **first modification** is to replace Eq. 3 with

$$\nabla_{\mathcal{C}_t} \log q_{\sigma}(\mathcal{C}_t|\mathcal{C}_0) = \sum_{i=1}^{n} \frac{1}{\mathrm{degree}_i} \sum_{j \in N(i)} \frac{\partial d_{ij}^{(t)}}{\partial \mathcal{C}_t} \nabla_{d_{ij}^{(t)}} \log q_{0t}(d_t|d_0) \tag{5}$$

where $\mathrm{degree}_i$ is the degree of node $i$. The **second modification** is to add a multiplier "scale", $k_{\mathbf{s}_{\boldsymbol{\theta}}}$ to Eq. 4:

$$\mathcal{C}_t \approx \mathcal{C}_s + k_{\mathbf{s}_{\boldsymbol{\theta}}}(d_s, s, t)\,[\sigma(t) - \sigma(s)]\,\mathbf{s}_{\boldsymbol{\theta}}(d_s, s) \tag{6}$$

The reason for the first and second modifications is detailed in Sec. 4.1 and Sec. 4.2. In our empirical investigation, we substantiate that our solver is capable of producing high-quality samples when contrasted with the approach of sampling through thousands of iterations of Langevin dynamics.

## 4.1 DIFFERENTIAL FORM OF CONFORMATION SCORE

We examine the relationship existing between the manifold of pairwise distances and conformation. We define the manifold of pairwise distance matrices or adjacent matrices to be $M$ and the manifold of the SE(3)-invariant conformation coordinates to be $N$. Consider mappings defined in Fig 1. By definition, we have

$$\nabla_{\mathcal{C}_t} \log q_{0t}(\mathcal{C}_t|\mathcal{C}_0) = \mathrm{d}\varphi_{d_t}(\nabla_{d_t} \log q_{0t}(d_t|d_0)) \tag{7}$$

**Computation of** $\mathrm{d}\varphi$. For fixed $t$, we rewrite $\tilde{d} := d_t, d := d_0$ and $p_{\sigma}(\tilde{d} \mid d) := q_{0t}(d_t \mid d_0)$. Consider the mapping $f : T_{\tilde{d}}M \to N$. We can write $f = \pi_N \circ \mathrm{d}\varphi = \varphi \circ \pi_M$. Since we use a $n \times 3$ matrix to embed $\mathcal{C} \in N$. In such an embedding, we can choose $\pi_N$ to be an

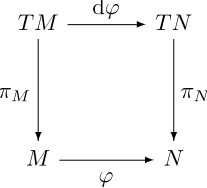

Figure 1: Mappings between the manifolds of pairwise distances $M$ and SE(3)-invariant conformations $N$. A more detailed introduction can be found in Appendix C.

identical mapping and we have $\mathrm{d}\varphi_{\tilde{d}} = \varphi \circ \pi_{M,\tilde{d}}$. The mapping $\varphi : M \to N$ is a function that maps the adjacent matrix to a conformation and $\pi_M$ can be chosen to be a generalized multidimensional scaling (GMD), i.e. given $\hat{d} = \tilde{d} + \sum_{i,j} \alpha_{ij} e_{ij} \in T_{\tilde{d}}M$, we define $\pi_{M,\tilde{d}}(\hat{d})_{ij} = \|\hat{\mathbf{x}}_i - \hat{\mathbf{x}}_j\|$, where $\hat{\mathbf{x}}_1, \ldots, \hat{\mathbf{x}}_n = \operatorname*{argmin}_{\hat{\mathbf{x}}_1, \ldots, \hat{\mathbf{x}}_n} \sum_{i<j} \left( \|\hat{\mathbf{x}}_i - \hat{\mathbf{x}}_j\| - \hat{d}_{ij} \right)^2$. Then,

$$\mathrm{d}\varphi_{\tilde{d}}(\hat{d}) = \varphi \circ \pi_{M,\tilde{d}}(\hat{d}) = \operatorname*{argmin}_{\hat{\mathbf{x}}_1, \ldots, \hat{\mathbf{x}}_M} \sum_{i<j} \left( \|\hat{\mathbf{x}}_i - \hat{\mathbf{x}}_j\| - \hat{d}_{ij} \right)^2. \tag{8}$$

We approximate the solution of the above optimization problem as

$$\mathrm{d}\varphi_{\tilde{d}}(\hat{d}) = \mathrm{d}\varphi_{\tilde{d}}(\tilde{d} + \sum_{i,j} \alpha_{ij} e_{ij}) \approx \tilde{\mathcal{C}} + \sum_{i,j} \frac{\alpha_{ij}}{2(n-1)} \frac{\partial \tilde{d}_{ij}}{\partial \tilde{\mathcal{C}}} \tag{9}$$

**Theorem 1.** *Consider the optimization problem $f(\hat{d}) := \min_{\hat{\mathbf{x}}_1, \ldots, \hat{\mathbf{x}}_M} \sum_{i<j} \left( \|\hat{\mathbf{x}}_i - \hat{\mathbf{x}}_j\| - \hat{d}_{ij} \right)^2$, where $\hat{d} = \tilde{d} + \delta e_{uv}$. The optimal value $f(\hat{d})$ approximated by $\hat{x} = \mathrm{d}\varphi_{\tilde{d}}(\hat{d})$ from Eq. 9 is bounded by $\frac{2n^2+n-1}{2(n-1)^2} \delta^2$ (See Appendix D.2 for the formal proof). Hence, approximation error does not explode with the increase of node numbers.*

Then, by Eq. 9 and the linearity of $\mathrm{d}\varphi_{\tilde{d}}$, we have

$$\nabla_{\tilde{\mathcal{C}}} \log p_\sigma(\tilde{\mathcal{C}}|\mathcal{C}) = \mathrm{d}\varphi_{\tilde{d}}(\nabla_{\tilde{d}} \log p_\sigma(\tilde{d}|d)) \approx \sum_{i,j} \frac{\nabla_{\tilde{d}_{ij}} \log p_\sigma(\tilde{d}|d)}{2(n-1)} \frac{\partial \tilde{d}_{ij}}{\partial \tilde{\mathcal{C}}} \tag{10}$$

Detailed intermediate steps and the reason for such an approximation, as well as approximation error bound and tangent space assumptions, are detailed in Appendix D. Since we usually consider partially connected conformations, the pairwise distance matrix is sparse. We thus modify Eq. 10 to

$$\nabla_{\mathcal{C}_t} \log q_{0t}(\mathcal{C}_t|\mathcal{C}_0) \approx \sum_{i=1}^n \frac{1}{\mathrm{degree}_i} \sum_{j \in N(i)} \frac{\partial d_{ij}^{(t)}}{\partial \mathcal{C}_t} \nabla_{d_{ij}^{(t)}} \log q_{0t}(d_t|d_0), \tag{11}$$

where $\mathrm{degree}_i$ denotes the degree of node $i$. By completing the sparse conformation into a fully connected conformation, Eq. 11 is reduced to Eq. 10. Various methods have been proposed to compute $\nabla_{d_t} \log q_{0t}(d_t|d_0)$ and details can be seen in Appendix E. Using these methods, we can train the model $\mathbf{s}_{\boldsymbol{\theta}}(d_t, t) \approx -\sigma_t \nabla_{\mathcal{C}_t} \log q_{0t}(\mathcal{C}_t|\mathcal{C}_0)$.

## 4.2 MODELING OF REVERSED FLOW

As stated in Eq. 2, the reverse flow of conformations is

$$\frac{\partial \mathcal{C}_t}{\partial t} = -\frac{1}{2} \frac{\mathrm{d}\sigma_t^2}{\mathrm{d}t} \nabla_{\mathcal{C}_t} \log q_{0t}(\mathcal{C}_t \mid \mathcal{C}_0) = -\frac{1}{2} \frac{\mathrm{d}\sigma_t^2}{\mathrm{d}t} \mathrm{d}\varphi_{d_t}(\nabla_{d_t} \log q_{0t}(d_t|d_0)) \tag{12}$$

where we assume $\mathcal{C}_t + \nabla_{\mathcal{C}_t} \log q_{0t}(\mathcal{C}_t \mid \mathcal{C}_0) \in T_{\mathcal{C}_t}N, d_t + \nabla_{d_t} \log q_{0t}(d_t \mid d_0) \in T_{d_t}M$. We can discretize the above ODE (Lu et al., 2022a) (see Appendix B) and suppose that we have a model $\mathbf{s}_{\boldsymbol{\theta}}(d_t, t) = -\sigma_t \nabla_{\mathcal{C}_t} \log q_{0t}(\mathcal{C}_t|\mathcal{C}_0) + \boldsymbol{\epsilon}(d_t, t)$, where $\boldsymbol{\epsilon}(d_t, t)$ is the prediction error, then we have

$$\mathcal{C}_t \approx \mathcal{C}_s - \sigma_s \sigma'(s) \mathrm{d}\varphi \left( \nabla_{d_s} \log q_{0s}(d_s) \right) \tag{13a}$$
$$\approx \mathcal{C}_s + [\sigma(t) - \sigma(s)] \mathbf{s}_{\boldsymbol{\theta}}(d_s, s) - [\sigma(t) - \sigma(s)] \boldsymbol{\epsilon}(d_s, s) \tag{13b}$$
$$:= \mathcal{C}_s + [\sigma(t) - \sigma(s)] \mathbf{s}_{\boldsymbol{\theta}}(d_s, s) + \bar{\boldsymbol{\epsilon}}(d_s, s, t) \tag{13c}$$

and intermediate time steps are uniformly distribution between $T$ and $0$. The term $\bar{\epsilon}(d_s, s, t)$ can be seen as an addtional noise injected to $\mathcal{C}_s$. Hence, after one iteration of the denoising process, the obtained $\tilde{C}_t$ should be $\tilde{C}_{t+\lambda(s-t)}$ for some $\lambda \in (0, 1)$. This motivates us to choose a larger drift of the conformation score. Hence, we introduce a multiplier "scale" $k_{\mathbf{s}_{\boldsymbol{\theta}}}(d_s, s, t) > 1$ as an addition term to remedy the model's prediction errors, and the iteration rule becomes

$$\mathcal{C}_t \approx \mathcal{C}_s + k_{\mathbf{s}_{\boldsymbol{\theta}}}(d_s, s, t) \left[\sigma(t) - \sigma(s)\right] \mathbf{s}_{\boldsymbol{\theta}}(d_s, s) \tag{14}$$

To find scale $k_{\mathbf{s}_{\boldsymbol{\theta}}}$, we further assume $k_{\mathbf{s}_{\boldsymbol{\theta}}}(d_s, s, t) \approx k_{\mathbf{s}_{\boldsymbol{\theta}}}(p_{\text{data}})$, i.e., for each dataset, we can find a hyper-parameter to approximate $k_{\mathbf{s}_{\boldsymbol{\theta}}}(d_s, s, t)$ for all $d_t \sim q_{0t}(d_t)$. From the experimental results, we find that the magnitude of $k_{\mathbf{s}_{\boldsymbol{\theta}}}$ increases along with the increase of the model's prediction error, while other factors such as node number and node degree do not have a significant influence on the choice of $k_{\mathbf{s}_{\boldsymbol{\theta}}}$. Details of the experiments and the results can be seen in Sec. 5.1.

# 5 EXPERIMENTS

We first examine the influential factors associated with the newly introduced hyper-parameter "scale" $k_{\mathbf{s}_{\boldsymbol{\theta}}}$ in Sec. 5.1. Then we evaluate our proposed accelerated sampling method through a comparative analysis with the baseline sampling method in Sec. 5.3 and conduct the hyper-parameter analysis for these two sampling method in Sec. 5.4. Finally, we assess our method's efficacy when the number of iterations in the sampling process is further reduced in Sec. 5.5. Appendix G provides additional experiments, and visualization of sampling process of our method can be found in Appendix H.

## 5.1 MODEL ERROR $\bar{\epsilon}$ AND SCALE $k_{\mathbf{s}_{\boldsymbol{\theta}}}$

We develop toy datasets $\{Q_i\}_{i=1}^n$ to investigate the relationship between the prediction error $\bar{\epsilon}$ and scale $k_{\mathbf{s}_{\boldsymbol{\theta}}}$ and each dataset only contains a single sample, i.e., $Q_i = \{\mathcal{C}_0^{(i)}\}$. For each dataset, we denote the sigma scheduler as $\{\sigma_t\}_{t=1}^T$ and set the forward diffusion process to be $q_{0t}(\mathcal{C}_t \mid \mathcal{C}_0) = \mathcal{N}(\mathcal{C}_t \mid \mathcal{C}_0, \sigma_t^2 \boldsymbol{I})$ and suppose we have a model $\mathbf{s}_{\boldsymbol{\theta}}^{Q_i}(d_t, \delta) = \text{Norm}_{\text{std}}\left(d\varphi\left((-d_t + d_0) + \boldsymbol{\epsilon}^d(\delta, t)\right)\right)$, where $d_t$ is the pairwise distance of the conformation coordinates at time $t$ and $\boldsymbol{\epsilon}^d(\delta, t)$ is the noise level term controlled by a hyper-parameter $\delta$. We assume $\mathbf{s}_{\boldsymbol{\theta}}^{Q_i}(d_t, \delta) \approx -\sigma_t \nabla_{\mathcal{C}_t} \log q_{0t}(\mathcal{C}_t|\mathcal{C}_0)$ when $\delta$ is small. Detailed reasons for such a definition and corresponding settings can be found in Appendix F.1. To generate samples, we sample random noise $\tilde{\mathcal{C}}_T \sim \mathcal{N}(\mathbf{0}, \sigma_T^2 \boldsymbol{I})$ and apply

$$\tilde{\mathcal{C}}_t \approx \tilde{\mathcal{C}}_s + k_{\mathbf{s}_{\boldsymbol{\theta}}}[\sigma(t) - \sigma(s)]\mathbf{s}_{\boldsymbol{\theta}}(d_s, s, \delta) \tag{15}$$

to generate $\tilde{\mathcal{C}}_0$. If $\|\tilde{d}_t - d_0\|_{\infty} < h$ for some predefined threshold $h > 0$, we say that the reverse process converges at $t$ and define

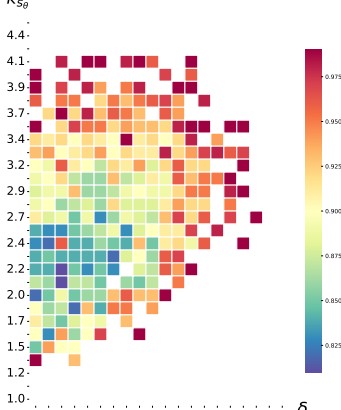

Figure 2: Relations between the prediction error $\delta$ and scale $k_{\mathbf{s}_{\boldsymbol{\theta}}}$. Grid color indicates the convergent time.

the minimal $1 - t/T$ to be the convergent time. We aim to find the convergent time under different noise levels $\delta$ and scale $k_{\mathbf{s}_{\boldsymbol{\theta}}}$. We grid-search the above two parameters and visualize the convergence of the model. The results can be seen in Fig. 2 and the color of each grid represents the convergent time. Grids with no color imply that under such noise level $\delta$ and model error $k_{\mathbf{s}_{\boldsymbol{\theta}}}$, the model diverges. We can see a positive correlation between the model's error and the scale, which matches the hypothesis in Eq. 13. Other ablation studies show that other factors including node number and node degree do not have a strong impact on the choice of $k_{\mathbf{s}_{\boldsymbol{\theta}}}$. Detailed analysis is in Appendix F.2.

## 5.2 EXPERIMENT SETUP

We firstly retrain SDDiff and GeoDiff models after the modifying their conformation score as introduced in Equation 5. Comprehensive training specifications can be found in Appendix G.1. It's important to note that the results presented here are derived from the modified models.

Table 1: Comparison results between LD sampling and our fast sampling method. Note that GeoDiff and SDDiff here refer to the revised ones with modified score estimation in Eq. 5. Higher values for COV indicate better results, while lower values for MAT are preferable. The reported time represents the average run time for sampling a single conformer.

| Dataset | Model | Sampling method | Recall | | | | Precision | | | | Time (s) |
|---|---|---|---|---|---|---|---|---|---|---|---|
| | | | COV-R(%) ↑ | | MAT-R(Å) ↓ | | COV-P(%) ↑ | | MAT-P(Å) ↓ | | |
| | | | Mean | Median | Mean | Median | Mean | Median | Mean | Median | |
| Drugs | GeoDiff | LD | 78.83 | 89.37 | 1.0422 | 1.0346 | **51.13** | **50.00** | **1.3143** | **1.2638** | 4.4506 |
| | | Ours | **84.06** | **94.39** | **0.9693** | **0.9608** | 49.09 | 49.04 | 1.3746 | 1.3303 | 0.0868 |
| | SDDiff | LD | 56.88 | 55.85 | 1.3318 | 1.2448 | **60.25** | **65.86** | **1.2619** | **1.1446** | 4.2392 |
| | | Ours | **70.69** | **76.68** | **1.0946** | **1.0801** | 48.83 | 48.01 | 1.6397 | 1.4478 | 0.0802 |
| QM9 | GeoDiff | LD | 88.79 | 93.00 | 0.3285 | 0.3249 | 50.84 | 48.30 | 0.6986 | 0.5027 | 2.4608 |
| | | Ours | **90.62** | **95.09** | **0.2427** | **0.2368** | **52.30** | **50.61** | **0.4714** | **0.4588** | 0.0471 |
| | SDDiff | LD | **90.56** | **95.58** | **0.2740** | **0.2693** | **52.74** | **50.26** | **0.6210** | **0.4660** | 2.2428 |
| | | Ours | 88.46 | 92.66 | 0.2920 | 0.2905 | 46.94 | 45.07 | 0.8293 | 0.6726 | 0.0459 |

**Dataset.** We employ two datasets, namely GEOM-Drugs and GEOM-QM9 (Axelrod & Gomez-Bombarelli, 2022) to validate the efficiency of our fast sampling method. The dataset split is from GeoDiff (Xu et al., 2020). In the GEOM-Drugs dataset's test set, we encounter a total of 14,324 conformers from 200 molecules, with an average of approximately 47 atoms per molecule. The GEOM-QM9 test set contains 24,143 conformers originating from 200 molecules, and each molecule has an average of around 20. In line with the previous work (Xu et al., 2020), we expand the generation of ground truth conformers to double their original quantity, resulting in more than 20k+ and 40k+ conformers being generated. Please refer to GeoDiff (Xu et al., 2022) for more information.

**Evaluation.** We adopt established evaluation metrics, namely COV (Coverage) and MAT (Matching), incorporating Recall (R) and Precision (P) aspects to assess the performance of sampling methods (Xu et al., 2020; Ganea et al., 2021). COV quantifies the proportion of ground truth conformers effectively matched by generated conformers, gauging diversity in the generated set. On the other hand, MAT measures the disparity between ground truth and generated conformers, complementing quality assessment. Furthermore, refined metrics COV-R and MAT-R place added emphasis on the comprehensiveness of the ground truth coverage, while COV-P and MAT-P are employed to gauge the precision and accuracy of the generated conformers. Detailed calculations are in Appendix G.1.

**Baseline sampling method.** To the best of our knowledge, no other fast conformation generation methods exist. Therefore, we compare our fast sampling approach with the conventional sampling method via Langevin dynamics (LD sampling) (Song & Ermon, 2019):

$$\mathcal{C}_{t-1} = \mathcal{C}_t + \alpha_t \nabla_{\mathcal{C}_t} \log p_\sigma(\mathcal{C}_t) + \sqrt{2\alpha_t}\mathbf{z}_{t-1}, \quad t = T, T-1, \ldots, 2, 1 \qquad (16)$$

where $\mathbf{z}_t \sim \mathcal{N}(\mathbf{0}, \mathbf{I})$ and $\alpha_t = h\sigma_t^2$. $h$ is the hyper-parameter referring to step size and $\sigma_t$ is the noise schedule in the forward diffusion process. We employ $T = 5000$ in the diffusion process, necessitating 5000 iterations in LD sampling.

## 5.3 COMPARISON WITH BASELINE SAMPLING METHOD

The LD sampling method needs thousands of steps to align with the diffusion process, whereas our fast sampling achieves the same goal with significantly fewer iterations. We compare our fast sampling method using 100 steps with LD sampling, and the results are shown in Tab. 1. Our evaluation criteria consist of eight metrics, and the results displayed in the table are obtained under hyper-parameter settings that ensure a well-balanced comparison among these evaluation criteria.

Tab. 1 shows that our fast sampling method can generate conformers of comparable quality to LD sampling while achieving a roughly 50-fold speed improvement. Overall, metrics related to Recall are satisfying, indicating that good diversity in conformers generated by our methods. However, there

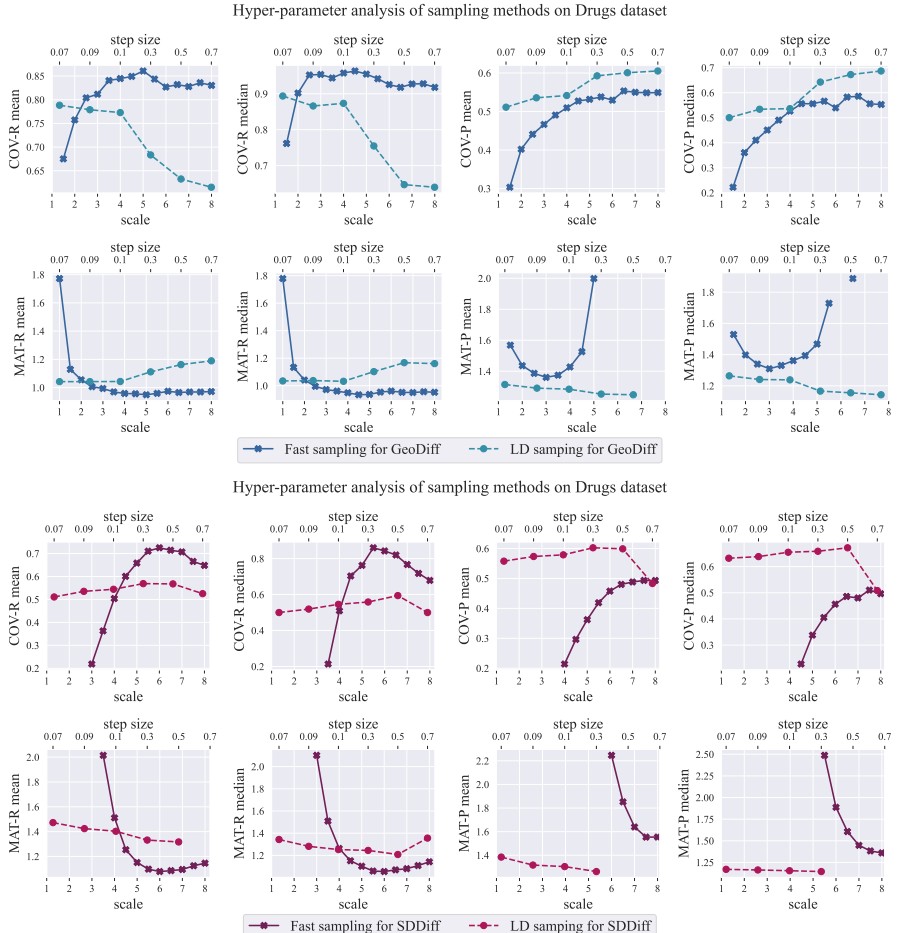

Figure 3: Impacts of scale $k_{\mathbf{s}_\theta}$ and step size $h$ in our fast sampling and LD sampling. Note that in the figures in the upper row, higher COV values signify superior performance, while in the figures below, the opposite holds for MAT. Metrics failing to meet the predefined thresholds (COV values lower than 0.2, MAT values higher than 2.5) have been intentionally omitted from the graphical representation for clarity and precision.

are more low-quality conformers generated, resulting in lower performance in terms of metrics under Precision.

## 5.4 HYPER-PARAMETER ANALYSIS

We introduce a hyper-parameter scale, denoted as $k_{\mathbf{s}_\theta}$ in Eq 14, to enable acceleration. In LD sampling, the hyper-parameter is step size $h$ in our setting. We illustrate the influence of these hyper-parameters on Drugs dataset in Fig. 3. Notably, certain data points revealed a significant underperformance are not depicted in the figure. A counterpart analysis on QM9 dataset is provided in Appendix G.2.

Fig. 3 shows that our method can obtain satisfactory metrics for Recall across most hyper-parameter values. However, metrics related to Precision consistently exhibit poorer results. Particularly, when higher scales are employed, resulting in significantly higher values for MAT-P. This is due to the deteriorating output of the network. We observe that part of the output occasionally start exploding from a certain sampling iteration especially under higher scale values, leading to some generated conformers becoming unstructured. This significantly impacts the P-series metrics but not the R-series metrics, as P-series metrics include all generated samples but the latter only considers conformers closely matching the ground truth. A similar phenomenon is also observed for LD sampling. When

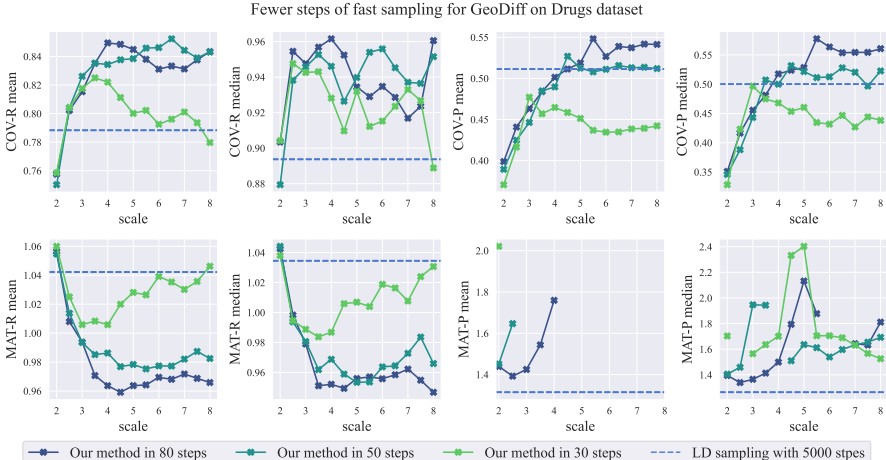

Figure 4: Results of our accelerated sampling technique applied to GeoDiff with a reduced number of steps. The dashed horizontal lines represent the results obtained through LD sampling. Data points with MAT metrics exceeding 2.5 have been omitted from the figure for clarity and precision.

the step size $h$ exceeds a certain range, P-series metrics quickly deteriorate. Addressing this issue may require improved network design in future research.

In the comparison between GeoDiff and SDDiff, we observe that when applied to SDDiff, our method demands a higher and more stringent scale $k_{\mathbf{s}_\theta}$. In a previous demonstration in Sec. 5.1, we demonstrate that scale is linked to modeling error. Despite SDDiff's more accurate distance score distribution approximation (Zhou et al., 2023), it leads to a significantly more complex learning objective compared to GeoDiff's Gaussian assumption. The presence of the random variable $\tilde{d}$, which can approach zero in the denominator (see Appendix E), poses a significant challenge to training. Consequently, the error in the trained models of SDDiff is more likely to be higher, thus requiring a higher value of scale within a narrower range.

## 5.5 SAMPLING IN FEWER STEPS

We evaluate the efficacy of our accelerated sampling method at further reduced steps. The results for the GeoDiff model on Drugs dataset are shown in Fig. 4, and more complementary experiments are detailed in Appendix G.3. As depicted in Fig. 4, our method exhibits noteworthy robustness when subjected to fewer steps. While performance gradually diminishes with decreasing step counts, it consistently maintains a commendable level of accuracy even under the constraint of only 30 steps. This adaptability to reduced step conditions underscores that our approach offers a compelling solution that strikes a commendable balance between speed and performance, indicating its considerable potential for real-world applications.

## 6 CONCLUSION

This study focuses on effective acceleration of diffusion-based generation in SE(3)-invariant space for molecular conformation generation. To this end, we first investigate the correlation between two manifolds regarding distances and coordinates utilizing an approximated differential operator, as well as rigorously validating this approximation through mathematical proofs and empirical experiments. Then, we alleviate the accumulation of approximation errors in the reverse diffusion process by introducing an additional hyper-parameter, scale. Empirical results support the validity of this remedial strategy, and detailed analysis provided insights into hyper-parameter selection. Building upon these findings, comparative investigations substantiate the effectiveness of our acceleration scheme. We posit that this study has the potential to expedite the sampling procedure in real-world applications, facilitating the practical deployment of diffusion models.

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

## A DIFFUSION SDEs AND SCORE MATCHING OBJECTIVE

Given a random variable $\mathbf{x}_0 \in \mathbb{R}^n$ following an unknown distribution $q_0$, diffusion probabilistic models (Song & Ermon, 2019; Ho et al., 2020a; Song et al., 2020b) define a forward process $\{\mathbf{x}_t\}_{t \in [0,T]}$ following a stochastic differential equation (SDE)

$$\mathrm{d}\mathbf{x}_t = f(t)\mathbf{x}_t\mathrm{d}t + g(t)\mathrm{d}\mathbf{w}_t, \quad \mathbf{x}_0 \sim q_0, \tag{17}$$

where $\mathbf{w}_t \in \mathbb{R}^n$ is the standard Brownian motion. Such a forward process has an equivalent reverse process starting from time $T$ to 0:

$$\mathrm{d}\mathbf{x}_t = \left[ f(t)\mathbf{x}_t - g^2(t)\nabla_{\mathbf{x}} \log q_t(\mathbf{x}_t) \right] \mathrm{d}t + g(t)\mathrm{d}\bar{\mathbf{w}}_t, \quad \mathbf{x}_T \sim q_{0T}(\mathbf{x}_T \mid \mathbf{x}_0), \tag{18}$$

and the marginal probability densities $\{q_{0t}(\mathbf{x}_t)\}_{t=0}^T$ of the above SDE is the same as the following probability flow ODE (Song et al., 2020b):

$$\frac{\mathrm{d}\mathbf{x}_t}{\mathrm{d}t} = f(t)\mathbf{x}_t - \frac{1}{2}g^2(t)\nabla_{\mathbf{x}} \log q_t(\mathbf{x}_t), \quad \mathbf{x}_T \sim q_{0T}(\mathbf{x}_T \mid \mathbf{x}_0). \tag{19}$$

This implies that if we can sample from $q_{0T}(\mathbf{x}_T) \approx q_{0T}(\mathbf{x}_T \mid \mathbf{x}_0)$ and solve Eq. 19, then the obtained $\mathbf{x}_0$ follows the distribution of $q_0(\mathbf{x}_0)$. The only unknown terms in Eq. 19 are $q_{0T}(\mathbf{x}_T)$ and $\nabla_{\mathbf{x}} \log q_t(\mathbf{x}_t)$. By choosing some specific $f(t)$ and $g(t)$, the distribution $q_{0T}(\mathbf{x}_T \mid \mathbf{x}_0)$ converges to $q_T(\mathbf{x}_T)$ as $T \to \infty$ and $q_T$ is an easy distribution for sampling like Gaussian distribution. To model $\nabla_{\mathbf{x}} \log q_t(\mathbf{x}_t)$, we can train a score-based model $\mathbf{s}_{\boldsymbol{\theta}^*}(\mathbf{x}_t, t)$ s.t.

$$\boldsymbol{\theta}^* = \arg\min_{\boldsymbol{\theta}} \mathbb{E}_t \left\{ \lambda_t \mathbb{E}_{\mathbf{x}_0 \sim q_0(\mathbf{x}_0)} \mathbb{E}_{\mathbf{x}_t \mid \mathbf{x}_0} \left[ \|\mathbf{s}_{\boldsymbol{\theta}}(\mathbf{x}_t, t) - \nabla_{\mathbf{x}} \log q_{0t}(\mathbf{x}_t \mid \mathbf{x}_0)\|_2^2 \right] \right\}, \tag{20}$$

where $\lambda_t = \lambda(t) : [0, T] \to \mathbb{R}_{++}$ is a weighting function, $t \sim [0, T]$, . The obtained model $\mathbf{s}_{\boldsymbol{\theta}^*}(\mathbf{x}_t, t)$ equals $\nabla_{\mathbf{x}_t} \log q_t(\mathbf{x}_t)$ for almost all $\mathbf{x}_t$ and $t$ (Song et al., 2020b).

## B DIFFUSION REVERSE ODE AND ACCELERATION

DPM solver (Lu et al., 2022a) is a high-order solver for the reverse diffusion ODE. Given a reverse process

$$\frac{\mathrm{d}\mathbf{x}_t}{\mathrm{d}t} = f(t)\mathbf{x}_t + \frac{g^2(t)}{2\sigma_t}\mathbf{s}_{\boldsymbol{\theta}}(\mathbf{x}_t, t), \quad \mathbf{x}_T \sim \mathcal{N}\left(\mathbf{0}, \sigma_T^2\mathbf{I}\right) \tag{21}$$

where

$$f(t) = \frac{\mathrm{d}\log\alpha_t}{\mathrm{d}t}, \quad g^2(t) = \frac{\mathrm{d}\sigma_t^2}{\mathrm{d}t} - 2\frac{\mathrm{d}\log\alpha_t}{\mathrm{d}t}\sigma_t^2. \tag{22}$$

The exact solution of diffusion ODEs is

$$\mathbf{x}_t = \frac{\alpha_t}{\alpha_s}\mathbf{x}_s - \alpha_t \int_{\lambda_s}^{\lambda_t} e^{-\lambda}\hat{\mathbf{s}}_{\boldsymbol{\theta}}(\hat{\mathbf{x}}_\lambda, \lambda)\,\mathrm{d}\lambda, \tag{23}$$

where $\lambda_t = \log(\alpha_t/\sigma_t)$, $\hat{\mathbf{s}}_{\boldsymbol{\theta}}(\hat{\mathbf{x}}_\lambda, \lambda) = \mathbf{s}_{\boldsymbol{\theta}}\left(\mathbf{x}_{t_\lambda(\lambda)}, t_\lambda(\lambda)\right)$ and $t_\lambda(\cdot)$ is the inverse function of $\lambda(t) = \lambda_t$ and satisfy $t = t_\lambda(\lambda(t))$. If we apply a first-order Taylor expansion of $\hat{\mathbf{s}}_{\boldsymbol{\theta}}(\hat{\mathbf{x}}_\lambda, \lambda)$ w.r.t. $\lambda$ at $\lambda_s$, we have

$$\mathbf{x}_t = \frac{\alpha_t}{\alpha_s}\mathbf{x}_s - \alpha_t\mathbf{s}_{\boldsymbol{\theta}}(\mathbf{x}_s, s)\int_{\lambda_s}^{\lambda_t} e^{-\lambda}\mathrm{d}\lambda + \mathcal{O}\left((\lambda_t - \lambda_s)^2\right) \tag{24a}$$

$$= \frac{\alpha_t}{\alpha_s} \mathbf{x}_s - \sigma_t \left( e^{\lambda_t - \lambda_s} - 1 \right) \mathbf{s}_{\boldsymbol{\theta}} \left( \mathbf{x}_s, s \right) + \mathcal{O} \left( (\lambda_t - \lambda_s)^2 \right). \tag{24b}$$

In our case, we take $\alpha = 1$ and Eq. 24 becomes

$$\mathbf{x}_t \approx \mathbf{x}_s - \sigma_t \left( e^{\log \frac{\sigma_s}{\sigma_t}} - 1 \right) \mathbf{s}_{\boldsymbol{\theta}} \left( \mathbf{x}_s, s \right) \tag{25a}$$

$$= \mathbf{x}_s - (\sigma_t - \sigma_s) \mathbf{s}_{\boldsymbol{\theta}} \left( \mathbf{x}_s, s \right) \tag{25b}$$

## C  INTRODUCTION OF DIFFERENTIAL GEOMETRY

In differential geometry, we consider mappings between two manifolds. Suppose that $\varphi : M \to N$ is a smooth map between smooth manifolds. In our case, $M$ denotes the manifold of pairwise distances and $N$ denotes the manifold of SE(3)-invariant coordinates. Then, $\varphi$ maps a pairwise distances $\tilde{d}$ to a set of coordinates $\tilde{\mathcal{C}}$, i.e. $\varphi(\tilde{d}) = \tilde{\mathcal{C}}$.

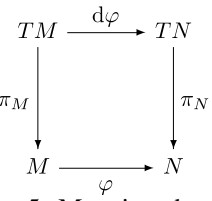

The differential of $\varphi$ at a point $\tilde{d}$, denoted as $\mathrm{d}\varphi_{\tilde{d}}$ is the best linear approximation of $\varphi$ near $\tilde{d}$. The differential is analogous to the total derivative in calculus. Mathematically speaking, the differential $\mathrm{d}\varphi$ is a linear mapping from the tangent space of $M$ at $\tilde{d}$ to the tangent space of $N$ at $\varphi(\tilde{d})$, which is $\mathrm{d}\varphi_{\tilde{d}} : T_{\tilde{d}}M \to T_{\varphi(\tilde{d})}N$.

Figure 5: Mappings between the manifolds of pairwise distances $M$ and SE(3)-invariant conformations $N$.

If tangent vectors are defined as equivalence classes of the curves $\gamma$ for which $\gamma(0) = \tilde{d}$ and we consider $\gamma(t)$ as the flow of the reversed diffusion process, then the differential is given by $\mathrm{d}\varphi_{\tilde{d}}(\gamma'(0)) = (\varphi \circ \gamma)'(0)$, which means that $\mathrm{d}\varphi_{\tilde{d}}$ maps the reversed flow of pairwise distance to the flow of the SE(3)-invariant coordinates. Hence, we write $\mathrm{d}\varphi_{\tilde{d}} \left( \tilde{d} + \nabla_{\tilde{d}} \log p_\sigma(\tilde{d} \mid d) \right) = \tilde{\mathcal{C}} + \nabla_{\tilde{\mathcal{C}}} \log p_\sigma(\tilde{\mathcal{C}} \mid \mathcal{C})$, where we assume that $\tilde{d} + \nabla_{\tilde{d}} \log p_\sigma(\tilde{\mathcal{C}} \mid d) \in TM$ and $\tilde{\mathcal{C}} + \nabla_{\tilde{\mathcal{C}}} \log p_\sigma(\tilde{\mathcal{C}} \mid \mathcal{C}) \in TN$. The above equation also implies that $\mathrm{d}\varphi_{\tilde{d}} \left( \nabla_{\tilde{d}} \log p_\sigma(\tilde{d} \mid d) \right) = \nabla_{\tilde{\mathcal{C}}} \log p_\sigma(\tilde{\mathcal{C}} \mid \mathcal{C})$ by linearity. Since we use an $n \times 3$ matrix to embed $\mathcal{C} \in \mathbb{R}^{n \times 3} / \mathrm{SE}(3)$. We can choose $\pi_N$ as an identical mapping.

## D  APPROXIMATION OF $\mathrm{d}\varphi(\hat{d})$

### D.1  APPROXIMATION FORMULA

Mathematically speaking, the dimension of the tangent space at every point of a connected manifold is the same as the dimension of the manifold itself. However, since there is no constraint on model's output $\nabla_{\tilde{d}} \log p_\sigma(\tilde{d} \mid d)$, we still consider the tangent space of pairwise distance manifold has the dimension of $n^2$ and we assume $\tilde{d} + \delta e_{uv} \in T_{\tilde{d}}M$ for all $\delta > 0$. We consider $\pi_N \circ \mathrm{d}\varphi(\hat{d}) = \mathrm{d}\varphi(\hat{d}) = \underset{\hat{\mathbf{x}}_1, \ldots, \hat{\mathbf{x}}_M}{\mathrm{argmin}} \sum_{i < j} \left( \|\hat{\mathbf{x}}_i - \hat{\mathbf{x}}_j\| - \hat{d}_{ij} \right)^2$. Given a pairwise distance $\tilde{d} \in M$ and associated coordinate $\tilde{\mathcal{C}} = [\tilde{\mathbf{x}}_1, \ldots, \tilde{\mathbf{x}}_n]^\top \in \mathbb{R}^{n \times 3}$, we first consider $\tilde{d} + \delta e_{uv} \in T_{\tilde{d}}M$ and approximate the solution of $\pi_N \circ \mathrm{d}\varphi(\tilde{d} + \delta e_{uv}) = [\hat{\mathbf{x}}_1, \ldots, \hat{\mathbf{x}}_n]^\top$ to be

$$\begin{cases} \hat{\mathbf{x}}_u = \tilde{\mathbf{x}}_u + \dfrac{\delta}{2(n-1)} \boldsymbol{\lambda}_{uv}, \\[2mm] \hat{\mathbf{x}}_v = \tilde{\mathbf{x}}_v - \dfrac{\delta}{2(n-1)} \boldsymbol{\lambda}_{uv}, \\[2mm] \hat{\mathbf{x}}_k = \tilde{\mathbf{x}}_k, \qquad k \neq u, v. \end{cases}$$

where $\boldsymbol{\lambda}_{uv} = \frac{\tilde{\mathbf{x}}_u - \tilde{\mathbf{x}}_v}{\|\tilde{\mathbf{x}}_u - \tilde{\mathbf{x}}_v\|}$.

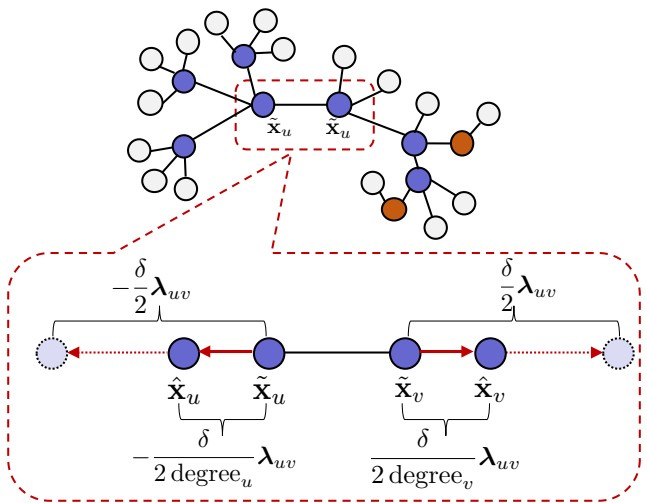

Figure 6: The illustration of approximating the solution of $\mathrm{d}\varphi(\tilde{d} + \delta e_{uv})$

Note that we can also write the solution of above as $\pi_N \circ \mathrm{d}\varphi(\tilde{d} + \delta e_{uv}) = \tilde{\mathcal{C}} + \frac{\delta}{2(n-1)} \frac{\partial \tilde{d}_{uv}}{\partial \tilde{\mathcal{C}}}$, since

$$
\left(\frac{\partial \tilde{d}_{uv}}{\partial \tilde{\mathcal{C}}}\right)_u = \left(\frac{\partial \|\tilde{\mathbf{x}}_u - \tilde{\mathbf{x}}_v\|}{\partial \tilde{\mathcal{C}}}\right)_u = \frac{\partial \|\tilde{\mathbf{x}}_u - \tilde{\mathbf{x}}_v\|}{\partial \tilde{\mathbf{x}}_u} = \frac{\tilde{\mathbf{x}}_u - \tilde{\mathbf{x}}_v}{\|\tilde{\mathbf{x}}_u - \tilde{\mathbf{x}}_v\|} = \boldsymbol{\lambda}_{uv} \tag{26a}
$$

$$
\left(\frac{\partial \tilde{d}_{uv}}{\partial \tilde{\mathcal{C}}}\right)_v = \left(\frac{\partial \|\tilde{\mathbf{x}}_u - \tilde{\mathbf{x}}_v\|}{\partial \tilde{\mathcal{C}}}\right)_v = \frac{\partial \|\tilde{\mathbf{x}}_u - \tilde{\mathbf{x}}_v\|}{\partial \tilde{\mathbf{x}}_v} = -\frac{\tilde{\mathbf{x}}_u - \tilde{\mathbf{x}}_v}{\|\tilde{\mathbf{x}}_u - \tilde{\mathbf{x}}_v\|} = -\boldsymbol{\lambda}_{uv} \tag{26b}
$$

$$
\left(\frac{\partial \tilde{d}_{uv}}{\partial \tilde{\mathcal{C}}}\right)_k = \left(\frac{\partial \|\tilde{\mathbf{x}}_u - \tilde{\mathbf{x}}_v\|}{\partial \tilde{\mathcal{C}}}\right)_k = \frac{\partial \|\tilde{\mathbf{x}}_u - \tilde{\mathbf{x}}_v\|}{\partial \tilde{\mathbf{x}}_k} = \mathbf{0}, \qquad k \neq u, v \tag{26c}
$$

Since $\mathrm{d}\varphi(\hat{d})$ is linear,

$$
\pi_N \circ \mathrm{d}\varphi(\hat{d}) = \mathrm{d}\varphi(\tilde{d} + \sum_{i,j} \alpha_{ij} e_{ij}) \tag{27a}
$$

$$
= (1 - n^2)\mathrm{d}\varphi(\tilde{d}) + \sum_{i,j} \mathrm{d}\varphi(\tilde{d} + \alpha_{ij} e_{ij}) \tag{27b}
$$

$$
= (1 - n^2)\tilde{\mathcal{C}} + n^2 \tilde{\mathcal{C}} + \sum_{i,j} \frac{\alpha_{ij}}{2(n-1)} \frac{\partial \tilde{d}_{ij}}{\partial \tilde{\mathcal{C}}} \tag{27c}
$$

$$
= \tilde{\mathcal{C}} + \sum_{i,j} \frac{\alpha_{ij}}{2(n-1)} \frac{\partial \tilde{d}_{ij}}{\partial \tilde{\mathcal{C}}}, \tag{27d}
$$

and

$$
\nabla_{\tilde{\mathcal{C}}} \log p_\sigma(\tilde{\mathcal{C}} | \mathcal{C}) = \mathrm{d}\varphi_{\tilde{d}}(\nabla_{\tilde{d}} \log p_\sigma(\tilde{d} | d)) \tag{28a}
$$

$$
= \mathrm{d}\varphi_{\tilde{d}}(\tilde{d} + \nabla_{\tilde{d}} \log p_\sigma(\tilde{d} | d)) - \mathrm{d}\varphi_{\tilde{d}}(\tilde{d}) \tag{28b}
$$

$$
= \tilde{\mathcal{C}} + \sum_{i,j} \frac{\nabla_{\tilde{d}_{ij}} \log p_\sigma(\tilde{d} | d)}{2(n-1)} \frac{\partial \tilde{d}_{ij}}{\partial \tilde{\mathcal{C}}} - \tilde{\mathcal{C}} \tag{28c}
$$

$$
= \sum_{i,j} \frac{\nabla_{\tilde{d}_{ij}} \log p_\sigma(\tilde{d} | d)}{2(n-1)} \frac{\partial \tilde{d}_{ij}}{\partial \tilde{\mathcal{C}}} \tag{28d}
$$

Finally, we have

$$\nabla_{\mathcal{C}_t} \log q_{0t}(\mathcal{C}_t) = \sum_{i,j} \frac{\nabla_{d_{ij}^{(t)}} \log q_{0t}(d_t \mid d_0)}{2(n-1)} \frac{\partial d_{ij}^{(t)}}{\partial \mathcal{C}_t} \tag{29}$$

## D.2 ERROR BOUND ANALYSIS

We compute the maximum objective function value of $\sum_{i<j} \left( \|\hat{\mathbf{x}}_i - \hat{\mathbf{x}}_j\| - \hat{d}_{ij} \right)^2$ under our approximation in the case of $\hat{d} = \tilde{d} + \delta e_{uv}$. Let $\sim$ denote the adjacent relation and $S_N(i) = \{(i,j) \mid \mathbf{x}_i \sim \mathbf{x}_j\}$ denotes the neighbors of node $i$, $S_\emptyset(i,j) = \{(p,q) \mid p \notin (i,j), q \notin (i,j)\}$ denote the set of nodes that are nonadjacent to $i$ and $j$, and $\text{degree}_i$ denote the degree of node $i$. We use $\hat{\mathcal{C}} = [\hat{\mathbf{x}}_1, \dots, \hat{\mathbf{x}}_n]$ to denote our approximated optimal solution, i.e. $\hat{\mathcal{C}} = \tilde{\mathcal{C}} + \sum_{i,j} \frac{\delta}{2(n-1)} \frac{\partial \tilde{d}_{ij}}{\partial \tilde{\mathcal{C}}}$. Formally, we have

$$\min_{\mathbf{x}_1,\dots,\mathbf{x}_n} \sum_{i<j} \left( \|\mathbf{x}_i - \mathbf{x}_j\| - \hat{d}_{ij} \right)^2 \tag{30a}$$

$$\leq \left( \|\hat{\mathbf{x}}_u - \hat{\mathbf{x}}_v\| - \hat{d}_{uv} \right)^2 + \sum_{(i,j) \in S_\emptyset(u,v)} \left( \|\hat{\mathbf{x}}_i - \hat{\mathbf{x}}_j\| - \hat{d}_{ij} \right)^2 \tag{30b}$$

$$+ \sum_{(i,j) \in S_N(u) \setminus \{(u,v)\}} \left( \|\hat{\mathbf{x}}_i - \hat{\mathbf{x}}_j\| - \hat{d}_{ij} \right)^2 + \sum_{(i,j) \in S_N(v) \setminus \{(u,v)\}} \left( \|\hat{\mathbf{x}}_i - \hat{\mathbf{x}}_j\| - \hat{d}_{ij} \right)^2 \tag{30c}$$

$$\leq \left( \left\| (\tilde{\mathbf{x}}_u + \frac{\delta}{2(n-1)} \boldsymbol{\lambda}_{uv}) - (\tilde{\mathbf{x}}_v - \frac{\delta}{2(n-1)} \boldsymbol{\lambda}_{uv}) \right\| - \tilde{d}_{uv} - \delta \right)^2 \tag{30d}$$

$$+ \sum_{(i,j) \in S_\emptyset(u,v)} \left( \|\tilde{\mathbf{x}}_i - \tilde{\mathbf{x}}_j\| - \tilde{d}_{ij} \right)^2 \tag{30e}$$

$$+ \sum_{(i,j) \in S_N(u) \setminus \{(u,v)\}} \left( \left\| (\tilde{\mathbf{x}}_u + \frac{\delta}{2(n-1)} \boldsymbol{\lambda}_{uv}) - \tilde{\mathbf{x}}_j \right\| - \tilde{d}_{uj} \right)^2 \tag{30f}$$

$$+ \sum_{(i,j) \in S_N(v) \setminus \{(u,v)\}} \left( \left\| (\tilde{\mathbf{x}}_v - \frac{\delta}{2(n-1)} \boldsymbol{\lambda}_{uv}) - \tilde{\mathbf{x}}_j \right\| - \tilde{d}_{vj} \right)^2 \tag{30g}$$

$$\leq \left( \|\tilde{\mathbf{x}}_u - \tilde{\mathbf{x}}_v\| - \tilde{d}_{uv} - \frac{\delta}{n-1} - \delta \right)^2 + 0 \tag{30h}$$

$$+ \max_{\pm(i,j)} \sum_{(i,j) \in S_N(u) \setminus \{(u,v)\}} \left( \|\tilde{\mathbf{x}}_u - \tilde{\mathbf{x}}_j\| \pm \left\| \frac{\delta}{2(n-1)} \boldsymbol{\lambda}_{uv} \right\| - \tilde{d}_{uj} \right)^2 \tag{30i}$$

$$+ \max_{\pm(i,j)} \sum_{(i,j) \in S_N(v) \setminus \{(u,v)\}} \left( \|\tilde{\mathbf{x}}_v - \tilde{\mathbf{x}}_j\| \pm \left\| \frac{\delta}{2(n-1)} \boldsymbol{\lambda}_{uv} \right\| - \tilde{d}_{vj} \right)^2 \tag{30j}$$

$$= \left( \frac{n}{n-1} \right)^2 \delta^2 + \sum_{(i,j) \in S_N(u) \setminus \{(u,v)\}} \left( \frac{\delta}{2(n-1)} \right)^2 + \sum_{(i,j) \in S_N(v) \setminus \{(u,v)\}} \left( \frac{\delta}{2(n-1)} \right)^2 \tag{30k}$$

$$= \left( \left( \frac{n}{n-1} \right)^2 + \frac{\text{degree}_u - 1}{4(n-1)^2} + \frac{\text{degree}_v - 1}{4(n-1)^2} \right) \delta^2 \tag{30l}$$

$$\leq \left( \left( \frac{n}{n-1} \right)^2 + \frac{n-1}{4(n-1)^2} + \frac{n-1}{4(n-1)^2} \right) \delta^2 \tag{30m}$$

$$= \frac{2n^2 + n - 1}{2(n-1)^2} \delta^2 \tag{30n}$$

Hence, the approximated optimal value is bounded above by $\frac{2n^2+n-1}{2(n-1)^2}\delta^2$.

### D.3 EXPERIMENTS OF APPROXIMATED OPTIMAL VALUE

**Experiment settings.** We randomly generate $\mathcal{C}_i \sim \mathcal{N}(\mathbf{0}_{n\times 3}, \boldsymbol{I})$ with $n$ nodes, where $n \sim$ Uniform($[10, 19]$). We compute the adjacent matrix $d$ of the obtained coordinates and randomly perturb the adjacent matrix to obtain $\hat{d} = d + \delta e_{uv}$. We aim to compare the magnitude of the optimal values computed under different algorithms or approximations. The optimal value is defined as the solution of the optimization problem $f(\hat{d}) = \min_{\hat{\mathbf{x}}_1,\ldots,\hat{\mathbf{x}}_n} \sum_{i<j} \left( \|\hat{\mathbf{x}}_i - \hat{\mathbf{x}}_j\| - \hat{d}_{ij} \right)^2$.

**Algorithms.** To our best knowledge, there is no simple algorithm for solving the metric MDS problem $\hat{\mathbf{x}}_1,\ldots,\hat{\mathbf{x}}_n = \operatorname*{argmin}_{\hat{\mathbf{x}}_1,\ldots,\hat{\mathbf{x}}_n} \sum_{i<j} \left( \|\hat{\mathbf{x}}_i - \hat{\mathbf{x}}_j\| - \hat{d}_{ij} \right)^2$. The usual gradient descent algorithm is inapplicable in such case since $\|\mathbf{x}\|$ is not differentiable at $\mathbf{x} = \mathbf{0}$. Usual convergence theorems for gradient methods are invalid under such cases and local minimum points do not need to satisfy the stationary equations (De Leeuw, 2005). Hence, we slightly modify the gradient descent process to

$$\hat{C}^{(t+1)} = \hat{C}^{(t)} + \nabla_{\hat{C}^{(t)}} \sum_{i<j} \left( \left\| \hat{\mathbf{x}}_i^{(t)} - \hat{\mathbf{x}}_j^{(t)} + \epsilon \right\| - \hat{d}_{ij} \right)^2, \tag{31}$$

for some sufficiently small $\epsilon > 0$. Then, we set $\hat{C}^{(0)} = \tilde{C} = [\tilde{\mathbf{x}}_1, \ldots, \tilde{\mathbf{x}}_n]$ as the initialized value and apply the gradient descent algorithm. We visualize the magnitude of the MDS objective function obtained at $\hat{\mathcal{C}}_{\text{approx}}, \hat{\mathcal{C}}_{\text{G}}, \hat{\mathcal{C}}_{\text{c-MDS}}$, computed from the proposed approximation (Eq. 9), gradient descent (Eq. 31) and the algorithm for the classic MDS problem, respectively. We also visualize the error bound proved in Eq. 30. The results can be seen in Fig 7. The algorithm for the classic MDS problem (c-MDS) is stated below (Wickelmaier, 2003):

- 1. Set up the squared proximity matrix $D = \left[ d_{ij}^2 \right]$

- 2. Apply double centering: $B = -\frac{1}{2}PDP$ using the centering matrix $P = I - \frac{1}{n}J_n$, where $n$ is the number of objects, $I$ is the $n \times n$ identity matrix, and $J_n$ is an $n \times n$ matrix of all ones.

- 3. Determine the 3 largest eigenvalues $\lambda_1, \lambda_2, \lambda_3$ and corresponding eigenvectors $e_1, e_2, e_3$ of $B$.

- 4. Now, $\mathcal{C} = E_3 \Lambda_3^{1/2}$, where $E_3$ is the matrix of 3 eigenvectors and $\Lambda_3$ is the diagonal matrix of 3 eigenvalues of $B$.

Metric MDS and c-MDS seek to find $\mathcal{C} = [\mathbf{x}_1, \ldots, \mathbf{x}_n]$ such that $\|\mathbf{x}_i - \mathbf{x}_j\| \approx d_{ij}$ but the optimal solution of c-MDS generally differs from the metric MDS.

**Experimental results.** We see that our approximation leads to a much smaller loss than the c-MDS's, and compared with the optimal value, the gap is acceptable. As there is still improvement for optimal values, this further suggests that applying a more refined projection operator can yield additional improvements in the model's performance (Zhou & Yu, 2023).

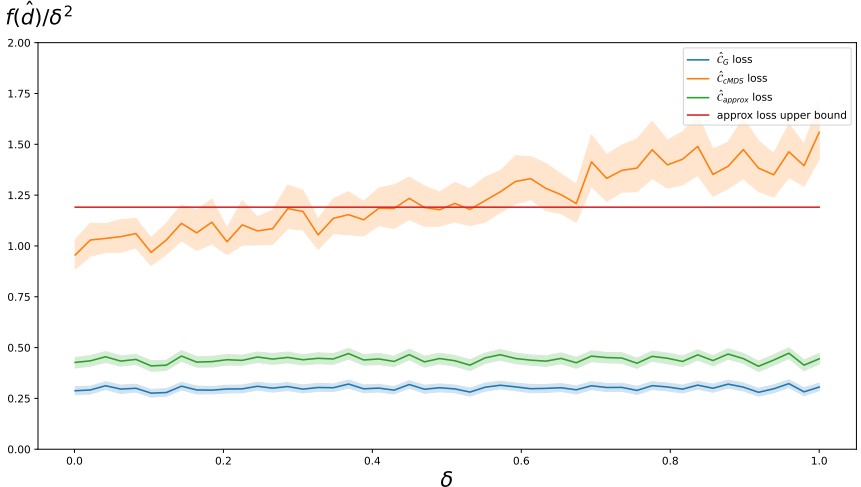

Figure 7: Optimization loss of MDS w.r.t. different optimization algorithm. $f(\hat{d})$ is the objective function of the optimization problem.

# E   COMPUTATION OF $\nabla_{d_t} \log q_{0t}(d_t|d_0)$.

We consider the score of $\nabla_{d_t} \log q_{0t}(d_t|d_0)$ where $d_{ij} = \|\mathbf{x}_i - \mathbf{x}_j\|$ when i.i.d. Gaussian noise is injected to conformation coordinates. GeoDiff (Xu et al., 2022) naively assumes that the perturbed distances follow a Gaussian distribution. In SDDiff (Zhou et al., 2023), authors proposed a shifting probability density to approximate the distribution of $q_{0t}(d_t|d_0)$. In this work, we consider both of the above two modeling methods and train two separate models with their own loss function:

$$\mathcal{L}\left(\boldsymbol{\theta}; \{\sigma_i\}_{t=1}^T\right) \triangleq \frac{1}{T}\sum_{t=1}^T \frac{1}{2} \mathbb{E}_{p_{\text{data}}(d_0)} \mathbb{E}_{q_{0t}(d_t|d_0)} \left\| \mathbf{s}_\theta(d_t, t) + \sigma_t \mathrm{d}\varphi_{d_t}\left(\nabla_{d_t}\log q_{0t}(d_t|d_0)\right)\right\|_2^2 \tag{32}$$

$$\text{GeoDiff: } \nabla_{d_t}\log q_{0t}(d_t|d_0) := -\frac{d_t - d_0}{\sigma_t^2} \tag{33}$$

$$\text{SDDiff: } \nabla_{d_t}\log q_{0t}(d_t|d_0) := \left(1 - e^{-\sigma_t/d_0}\right)\frac{8}{d_t} - 2\frac{d_t - d_0}{\sigma_t^2} \tag{34}$$

If a model $\mathbf{s}_\theta(d_t, t)$ minimizes the above loss, then $\mathbf{s}_\theta(d_t, t) \approx -\sigma_t \nabla_{\mathcal{C}_t}\log q_{0t}(\mathcal{C}_t|\mathcal{C}_0)$.

# F   FACTORS RELATED TO SCALE $k_{\mathbf{s}_\theta}$

In this section, we study factors that may influence the choice of scale $k_{\mathbf{s}_\theta}$. We mainly study factors including prediction error, node number, and node degree.

## F.1   SUPPLEMENTARY FOR POSITIVE CORRELATION BETWEEN PREDICTION ERROR AND SCALE $k_{\mathbf{s}_\theta}$

**Dataset settings.**   We develop datasets $Q_i = \{\mathcal{C}_0^{(i)}\}$ that only contain one conformation of 20 nodes. Each conformation is a 4-regular graph (so that SE(3)-invariant conformation and pairwise distance manifolds are surjective) and $\mathcal{C}_0^{(i)} \sim \mathcal{N}(\mathbf{0}, \boldsymbol{I}), \forall i = 1, \ldots, n$. Thus, the ground truth of the denoising process is fixed. The sigma scheduler $\{\sigma_t\}_{t=1}^T$ is chosen to be the same as that in the GeoDiff and we try to sample conformations in 100 steps.

**Model settings.**   During the denoising process, our model has the access to the ground truth $d_0$ and following GeoDiff's assumption of the distance distribution, we want to develop a model

$\mathbf{s}_{\boldsymbol{\theta}}(d_t, t) = \sigma_t \mathrm{d}\varphi \left(\nabla_{d_t} \log p_{0t}(d_t \mid d_0)\right) = \mathrm{d}\varphi \left(\frac{d_t - d_0}{\sigma_t}\right)$. As discussed in Eq. 13, we cannot access the accurate information about the denoising timestamp during the denoising process, hence, we assume that $\mathbf{s}_{\boldsymbol{\theta}}(d_t, t) \approx \mathrm{Norm}_{\mathrm{std}}\left(\mathrm{d}\varphi \left(d_t - d_0\right)\right)$. Such assumption comes from the fact that the standard deviation of the score matching ground truth $\sigma_t \mathrm{d}\varphi \left(\nabla_{d_t} \log p_{0t}(d_t \mid d_0)\right)$ approximately equals 1. Thus, the obtained model trends to output a score whose std equals 1. So, we force the output std of our model to be 1. Also, we use a hyperparameter $\delta$ to add prediction errors by letting $\mathbf{s}_{\boldsymbol{\theta}}(d_t, t, \delta) = \mathrm{Norm}_{\mathrm{std}}\left(\mathrm{d}\varphi \left(d_t - d_0\right) + \boldsymbol{\epsilon}^d(\delta, t)\right)$. But note that when $\delta = 0$, there are still prediction errors due to the approximated projection operator and the inaccurate distance distribution hypothesis. Finally, we define $\boldsymbol{\epsilon}^d(\delta, t) = 2(\mathrm{sigmoid}(\sigma_t \cdot \delta)\text{-}0.5)\delta\mathbf{z}$, where $\mathbf{z} \sim \mathcal{N}(\mathbf{0}, \boldsymbol{I})$.

**Detailed settings of convergence.** We find that even with access to the ground truth of $d_0$, our model cannot always converge to the ground truth at the end of the reverse process. Hence, we only consider the convergence for most samples (90% samples). Given a noise level $\delta$ and $k_{\mathbf{s}_\theta}$, we repeatedly apply the denoising process. If at time $t_0$, at least 90% samples have converged, then we say that under $\delta, k_{\mathbf{s}_\theta}$, the model converges at time $t_0$.

## F.2 REVERSE FLOW ANALYSIS

In the context of our investigation, we employ $\|d_t - d_0\|_\infty$ as a reduced-dimensional representation of the reversed flow phenomenon, where $d_t$ denotes the edge lengths of the graph at time $t$, $d_0$ denotes the ground truth edge lengths and $\|\cdot\|_\infty$ denotes the maximum difference. It is well-established that the selection of the scale $k_{\mathbf{s}_\theta}$ exerts a notable influence on the characteristics of the flow. Our primary objective is to systematically investigate whether while maintaining a constant scale parameter $k_{\mathbf{s}_\theta}$, other variables such as the number of nodes and node degrees exhibit substantial alterations in the reverse flow patterns. Should our findings indicate minimal variation in the reverse flow with respect to these aforementioned factors, it would enable us to posit that the choice of scale $k_{\mathbf{s}_\theta}$ is relatively independent of their influence. We use the same settings of the sigma scheduler and the model, as discussed in the Appendix. F.1 but develop different toy datasets that contain graphs of distinct node numbers and distinct node degrees.

**Dataset settings.** We develop two datasets with one containing a fixed node number of [10, 20, 50, 100], and each graph is a 4-regular graph and the other contains conformations of 20 nodes, and each graph is a $k$-regular graph, where $k = 4, 5, 6$.

**Experimental results.** We visualize the flow (represented by $\|d_t - d_0\|_\infty$) under different node numbers and node degrees and find that these two factors have a limited effect on the flow when the scale $k_{\mathbf{s}_\theta}$ is fixed. Results can be seen in Fig. 8a and 8b. We also visualize the reverse flow of 4-regular graphs with 20 nodes under different prediction errors. Results are shown in Fig. 8c. Compared with Fig. 8a and 8b, we can see that the flow under different settings of the model's error shows great difference and we conclude that prediction error is the main factor that affects the choice of scale $k_{\mathbf{s}_\theta}$.

## G EXPERIMENT EXTENSION

### G.1 SETTINGS

**Evaluation.** We use the COV and MAT metrics for Recall (R) and Precision (P) to evaluate the diversity and quality of generated conformers. These metrics are built on the root-mean-square-deviation (RMSD) of heavy atoms. COV can reflect the converge status of ground truth conformers, and MAT denotes the average minimum RMSD. The calculation for COV-R and MAT-R is:

$$\text{COV-R} = \frac{1}{|S_r|}\{\mathcal{C} \in S_r \mid \text{RMSD}(\mathcal{C}, \mathcal{C}') < \tau, \exists \mathcal{C}' \in S_g\}, \quad \text{MAT-R} = \frac{1}{|S_r|}\sum_{\mathcal{C}' \in S_g} \text{RMSD}(\mathcal{C}, \mathcal{C}')$$

where $S_g$ and $S_r$ denote generated and ground truth conformations. Sweeping $S_g$ and $S_r$, we obtain the COV-P and MAT-P. The MAT is calculated under RMSD threshold $\tau$. Following previous work, we set $\tau = 0.5\text{Å}$ for GEOM-QM9 and $\tau = 1.25\text{Å}$ for GEOM-Drugs.

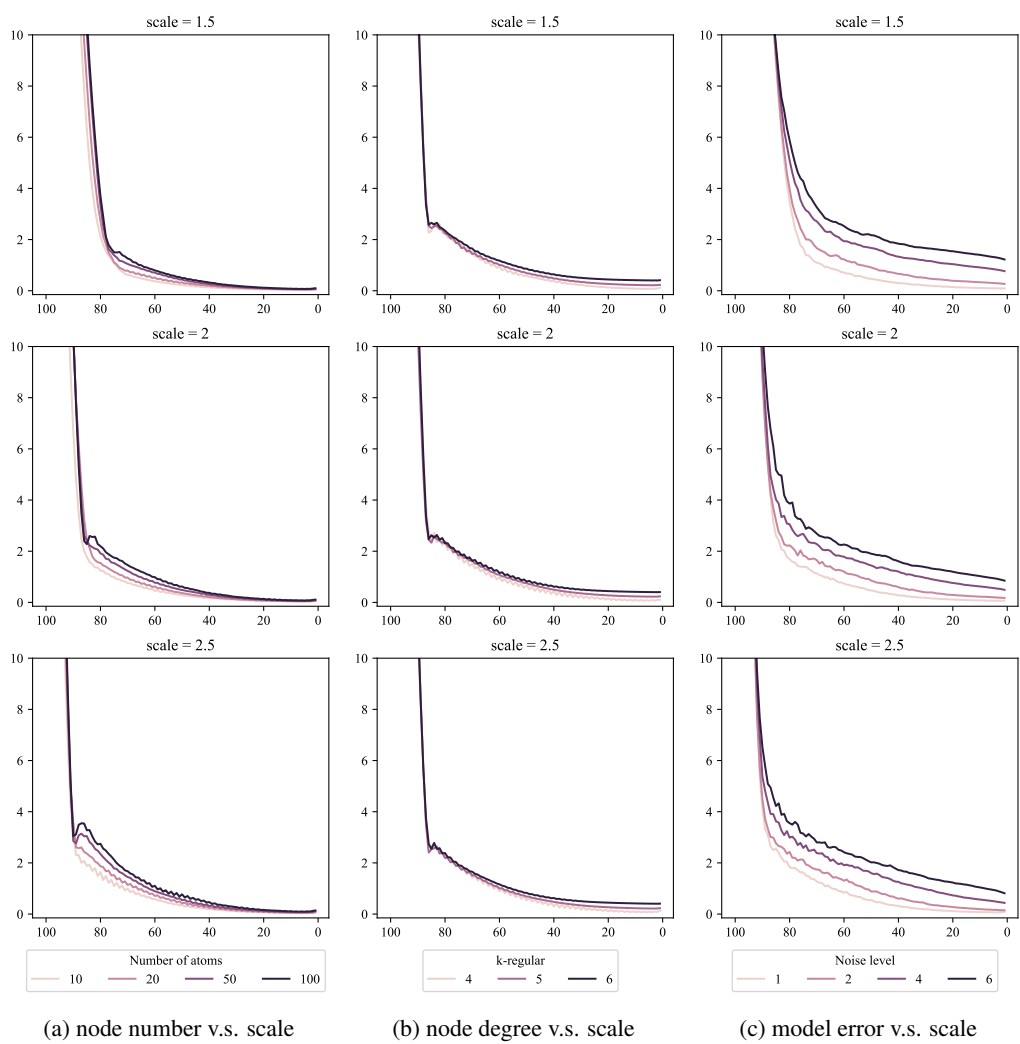

(a) node number v.s. scale      (b) node degree v.s. scale      (c) model error v.s. scale

Figure 8: Ablation study of factors that may affect the scale $k_{\mathbf{s}_\theta}$. The y-axis denotes the flow represented by $\|d_t - d_0\|_\infty$ and the x-axis denotes the denoising steps $t$. We can clearly see that the prediction error greatly affects the reverse flow.

**Training details** We adopt the same backbone network from GeoDiff. The backbone consists of local and global parts. The local part is GIN (Xu et al., 2018) and the global part is SchNet (Schütt et al., 2017). We trained the model on a single NVIDIA GeForce RTX 3090 GPU and Intel(R) Xeon(R) Silver 4210R CPU @ 2.40GHz CPU. We use the Adam optimizer for training, with a maximum of 1000 epochs. We set the learning rate to 0.001, and it decreases by a factor of 0.6 every 2 epochs. The batch size is 64. For SDDiff, the parameter only updates when loss satisfies two conditions: loss of the global part is lower than 0.75 on QM9 dataset or 2 on Drugs dataset, and total loss is lower than 10. We set the steps of the diffusion process to 5000, the noise scheduler to $\sigma_t = \sqrt{\frac{\bar{\alpha}_t}{1-\bar{\alpha}_t}}$ where $\bar{\alpha}_t = \prod_i^t (1 - \beta_i)$ and $\beta_t = \mathrm{sigmoid}(t)$. $t$ is uniformly selected in [1e-7, 2e-3], and the magnitude of $\sigma_t$ is in the range of 0 to around 12.

### G.2 MORE HYPER-PARAMETER ANALYSIS

In Sec. 5.4 we have investigated the influence of the hyper-parameter in LD sampling and our fast sampling. We conduct the same hyper-parameter analysis on QM9 dataset here. As shown in Fig. 9, we can draw a similar conclusion, that when $k_{\mathbf{s}_\theta}$ and $h$ are over a certain value, the P-series metrics

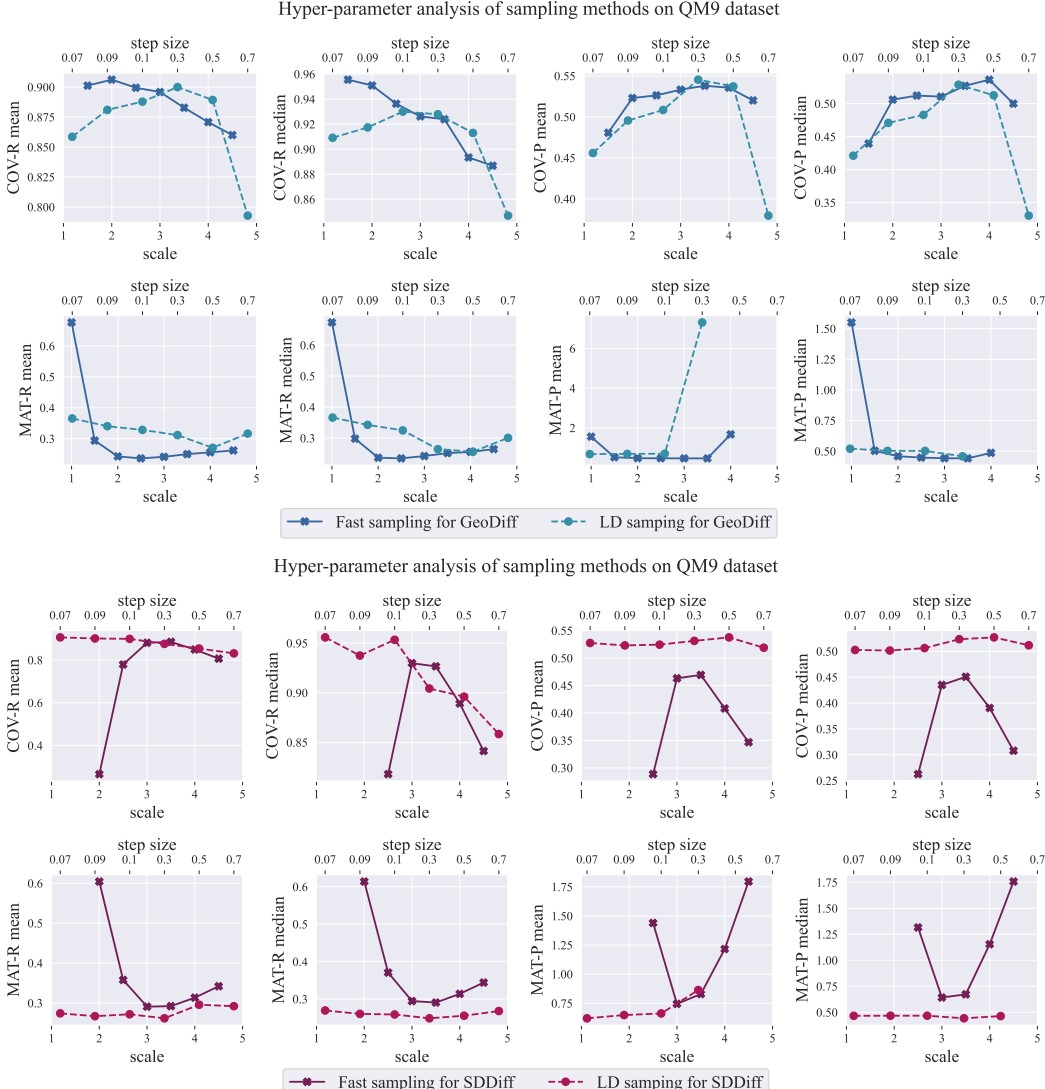

Figure 9: The influence of hyper-parameter, namely scale $k_{s_\theta}$ and step size $h$. Note that higher values of COV in the row above are preferable, whereas in the row below, higher MAT values are undesirable. Values falling below the defined thresholds (COV lower than 0.2 or MAT higher than 2.5) are disregarded and indicated as missing points in the figure.

become very poor. We think the reason is due to the low robustness of the network we use, details have been introduced in Sec. 5.4.

### G.3 MORE EXPERIMENTS IN FEWER STEPS

We have shown results on fewer steps for GeoDiff on Drugs dataset in Sec. 5.5 to illustrate that our methods have a certain level of robustness. Here we report the more experiments in fewer steps in Fig. 10.

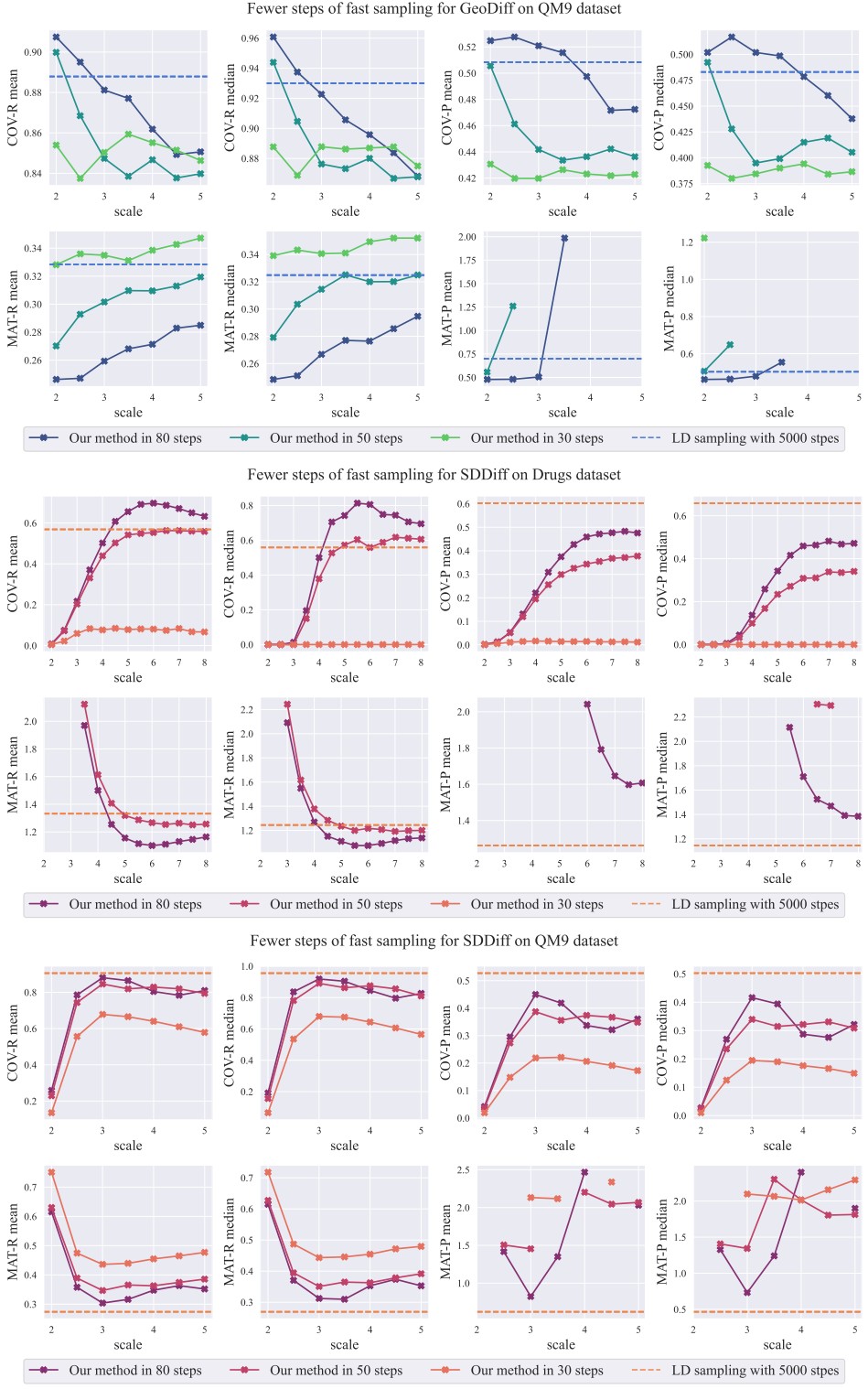

Figure 10: Results of our fast sampling method in fewer steps. The results of LD sampling in 5000 steps are shown as dashed horizontal lines. Points with poor MAT metric (higher than 2.5) are dropped in the figure.

# H  VISUALIZATION OF SAMPLING PROCESS

We visualize the generation process of an example conformer via our fast sampling process. As shown in Fig. 11, the conformer will determine the basic structure in very early steps, and the structure will undergo only minor adjustments in the subsequent majority of steps. This indicates that it is possible to obtain a faster sampling method.

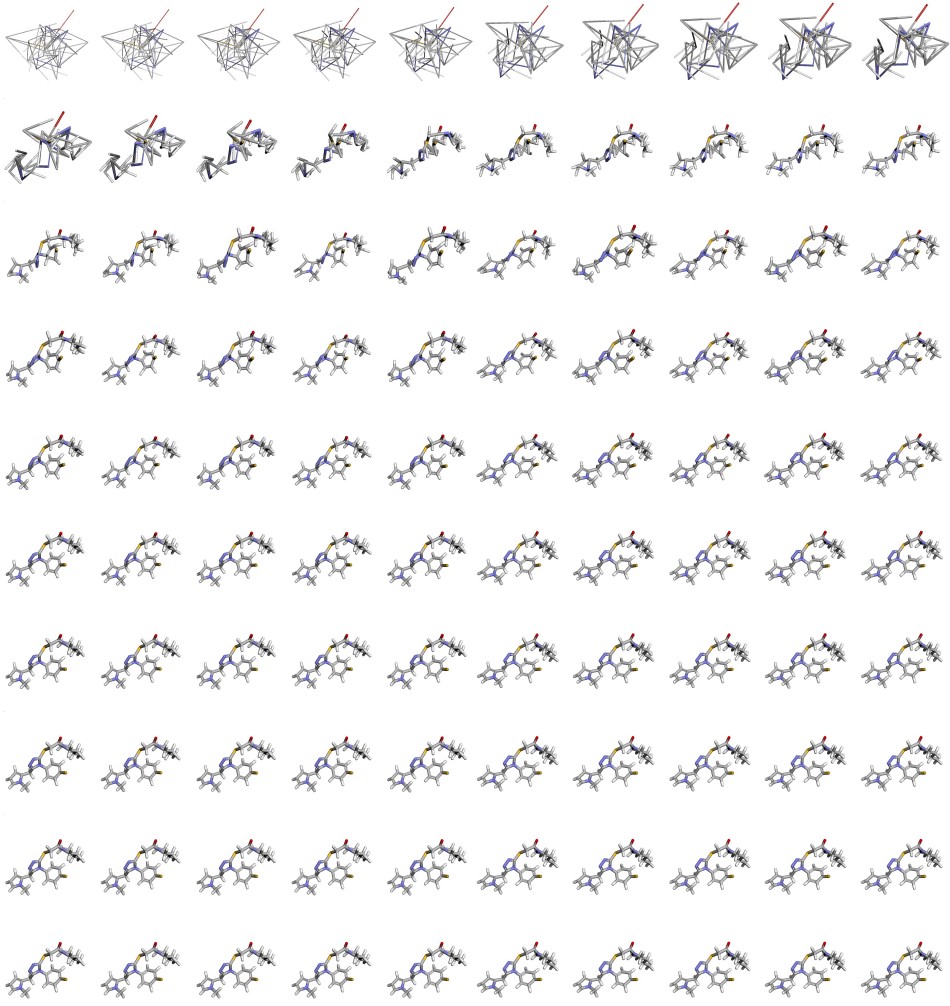

(a) Sampling process in 100 steps

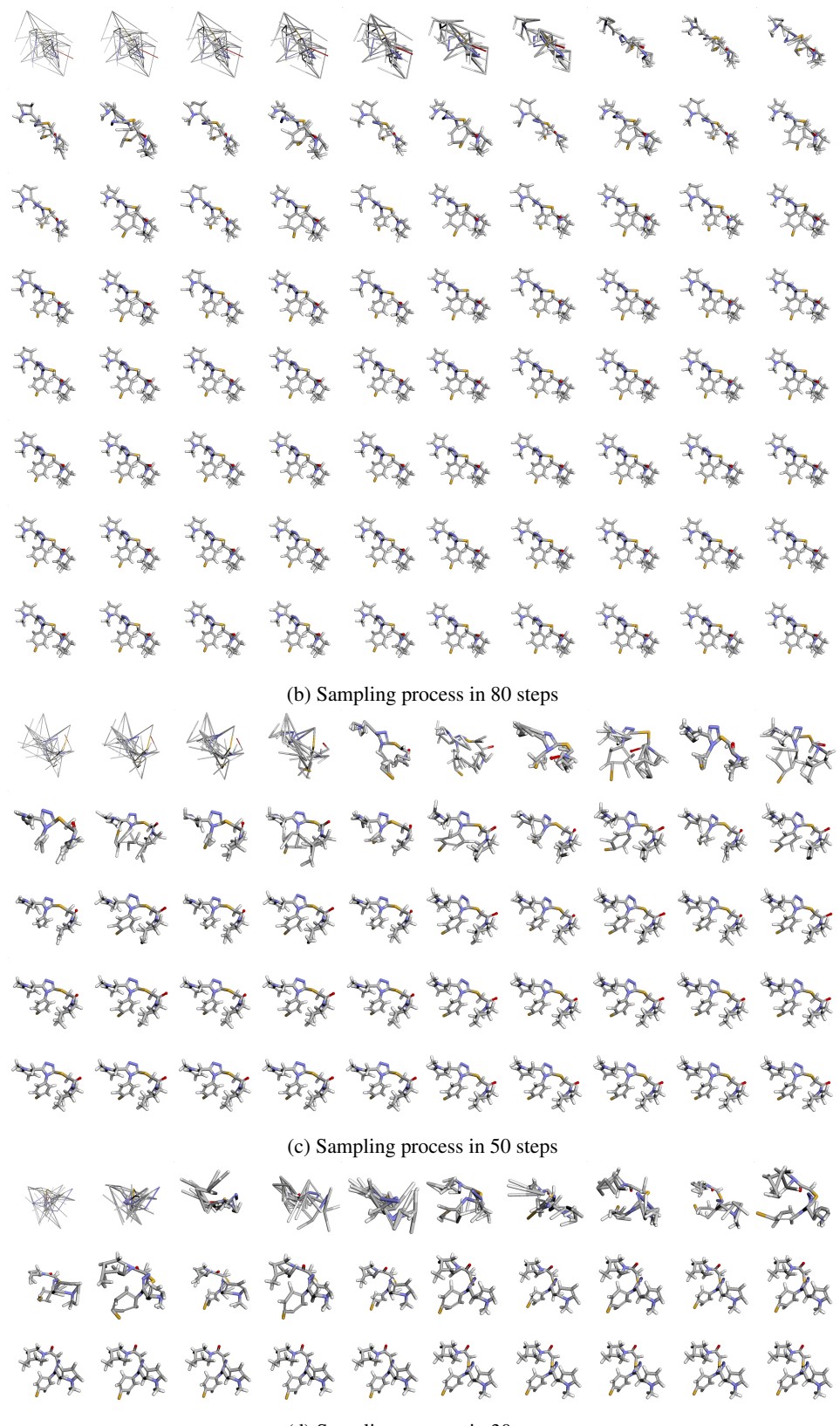

(b) Sampling process in 80 steps

(c) Sampling process in 50 steps

(d) Sampling process in 30 steps

Figure 11: Visualization of sampling process in different steps for GeoDiff on Drugs

