# OpenReview forum: "On Accelerating Diffusion-based Molecular Conformation Generation in SE(3)-invariant Space"
_ICLR.cc/2024/Conference — Submitted to ICLR 2024_

### Official Review · Reviewer_gijA · 2023-10-31

**Soundness:** 2 fair
**Presentation:** 2 fair
**Contribution:** 3 good
**Rating:** 5
**Confidence:** 4

**Summary:**

This paper proposes an acceleration method for diffusion-based molecular conformation generation models. The proposed method involves modifying the score-based model in coordinate space and applying multiplier to the discretization of the SDE. The first modification is motivated by an analysis of the relationship between scores in coordinate and distance spaces, while the second addresses score approximation errors. The proposed sampling technique is applied to two SE(3)-equivariant diffusion models, GeoDiff (Xu et al., 2022) and SDDiff (Zhou et al., 2023), and is compared to the annealed Langevin dynamics sampler. While the proposed sampler shows empirical advantages, the paper falls short in terms of justifying the modifications adequately, and the mathematical details are both lacking in rigor and difficult to follow. Without further elaboration on these details and their underlying motivation, the paper has yet to provide compelling evidence to demonstrate a substantial contribution.

**Strengths:**

Benchmarks on two conformer datasets (GEOM-Drugs and GEOM-QM9) demonstrate that the proposed method generates conformers of comparable quality and coverage to the reference method, but with a reduced computational time, requiring 50x–100x fewer time steps.

**Weaknesses:**

The argument presented in this paper relies on the assumption that the set of the valid distance matrices can be endowed with a manifold structure, as stated in Section 3.3: “The manifold of valid distance matrices is a proper sub-manifold of R_+^(n×n).” Subsequently, the pushforward map dφ is employed to map the score functions from distance space to coordinate space. However, the construction of a manifold structure for distance matrices is not trivial, considering that not all “distance matrices” are valid; they must satisfy conditions such as the triangle inequality. The authors should provide relevant references or mathematical construction to support the claims made in the paper.
	The differential geometry arguments in the paper lack clarity. For example, the use of dφ in this paper is more of a “proposal” than a well-defined mathematical construct. Additionally, while d ̃∈M and ∇_d ̃   log⁡〖p_σ (d ̃│d)∈T_d ̃  M〗, it is repeatedly “assumed” in the paper that d ̃+∇_d ̃   log⁡〖p_σ (d ̃│d)∈T_d ̃  M〗, even though the addition between elements of a manifold and its tangent space is not clearly defined.
	The rationale behind the factor of 1/2(n-1) in eq 9 and the unnumbered equation on page 14 is not clear. Since this factor is directly related to the construction of the first modification (eqs 5 and 11), it should be thoroughly explained and justified. It would be better to clarify whether this factor is based on the “rescaling” approximation illustrated in Figure 6.
	The implications of the error bound analysis in Theorem 1 are not clear. At least, the analysis does not support the claim that the proposed solution in eq 9 is “optimal”, as a naive choice of C ̂=C ̃ (the original coordinates) would yield a tighter upper bound of f(d ̂ )=δ^2<(2n^2+n-1)/(2(n-1)^2 ) δ^2 if n>1.
	In Section 4.2, the authors assume that the prediction error of the score-based model is a random error leading to an incomplete time step. However, the model error incurred during the score matching training procedure is more likely to be systematic rather than random, i.e., the model error would depend on the input distances and time steps. Since the following analysis in Section 5.1 assumes a random noise scheme, the authors would need to discuss how the results generalize to the trained score-based models with systematic errors.
	While the method generates conformers of comparable quality to the 5000-step LD sampler, it would be beneficial to demonstrate that the quality is superior to the LD sampler with equivalent time steps. It would be helpful to compare the results with the LD sampler with a similar (~100) or somewhat larger (500 or 1000?) number of time steps in Table 1.

**Questions:**

In Section 2, FrameDiff (Yim et al., 2023) performs diffusion on frame translations and frame orientations (rotations) rather than torsion angles as mentioned in the paper.
	On the top of page 5, symbols α_ij and e_ij are introduced without explanation. Also, it would be nice to provide a brief explanation or reference to the generalized multidimensional scaling.
	In the sentence just before eq 11, the meaning of the phrase “the pairwise distance matrix is sparse” is not clear. It might be better to reformulate the optimization problem in eq 8 by replacing the summation over all i < j pairs with connected (i, j) pairs.
	In Appendix G, the MAT-R metric should be corrected as follows:

MAT=1/|S_r |  ∑_(C∈S_r)▒min┬(C^'∈S_g )⁡RMSD(C,C^' )

---

> ### Author Response · Authors · 2023-11-19
>
> **Weakness**
> 1. **About manifold construct**. In this paper, we do not need to explicitly construct the manifold of the adjacent matrix and the SE(3)-invariant coordinates manifold. If one is interested in such a construction, one can easily construct the manifold of the adjacent matrix manifold by considering the spectral theorem. More specifically, for any feasible $D$, there corresponds to a Gram matrix $M \in \mathbb{R}^{n \times n}$ by $M_{ij} = \frac{1}{2} (D_{1j} + D_{i1} - D_{ij})$ and vice versa $D_{ij} = M_{ii} + M_{jj} - 2 M_{ij}$ and the matrix $M$ has a specific structure
> \begin{equation}
>     M=\left(\begin{array}{ll}
>     0 & \mathbf{0}^{\top} \\
>     \mathbf{0} & L
>     \end{array}\right)
> \end{equation}
> with $L \in \mathbb{R}^{(n-1) \times (n-1)}$ being symmetric and positive semi-definite and $\operatorname{rank}(L)=3$. Then, we define the mapping from $L$ to $D$ by $h$. One can define the manifold of the adjacent matrix to be $M = \{h(XX^T) \mid X \in \mathbb{R}^{(n-1) \times 3}\}$. To construt a SE(3)-invariant coordinate manifold, we pick any $\mathcal{C} \in \mathbb{R}^{n \times 3}$, then define $N_{\mathcal{C}} = \{ \hat{\mathcal{C}} = [\hat{\mathbf{x}}_1, \ldots, \hat{\mathbf{x}}_n] \mid \hat{\mathbf{x}}_i = \mathbf{x}_i + \sum_{j=1}^n \alpha_{ij} \frac{\mathbf{x}_i - \mathbf{x}_j}{\| \mathbf{x}_i - \mathbf{x}_j \|}, \alpha_{ij} = \alpha_{ji}, \alpha_{ii} = 0, \alpha_{ij} \in \mathbb{R} \}$. In this paper, we do not need to explicitly define the above two manifolds, hence, we do not consider such constructions.
>
> 2. **About manifold assumption.** We assume that the tangent space of the adjacent matrix manifold is $\mathbb{R}^{n \times n}$. Such an assumption is certainly wrong. However, since we did not apply any constraints on the model's output. When our model predicts the tangent vector of the tangent space of the distance manifold, the output may not necessarily lay in the tangent space. This pushes us to consider the tangent space as the $\mathbb{R}^{n \times n}$. The above argument for why we assume the tangent space is $\mathbb{R}^{n \times n}$ can also be found in Appendix D.1.
>
> 3. **About factor** $\frac{1}{2(n-1)}$. This factor is composed of two parts. One is $2$, which comes from the bipartition of the distance change, as illustrated in Fig. 6 by $-\frac{\delta}{2} \boldsymbol{\lambda}_{uv}$ and $\frac{\delta}{2} \boldsymbol{\lambda}_{uv}$. The other is $n-1$, which is the node degree in a fully connected graph.
>
> 4. **About error bound analysis.** The error-bound analysis aims to show that when the node number increases, the approximation error is still bounded. However, using GeoDiff's modeling of the conformation score, the error bound explodes when $n$ becomes sufficiently large. For the naive choice of the original coordinates, we can show that by experimental results on the toy dataset, the proposed approximation has a smaller average error compared with using the original coordinates.
>
> 5. **About prediction error.** Since we have no enough information about the structure of the prediction error, we can only apply a common assumption that the prediction error follows a Gaussian distribution.
>
> 6. **About abalation study.** This is a fantastic point. We reduced the $\sigma$ sequence from 5000 to 100 equally spaced points utilizing linear interpolation, the initial and final values from the original set are included. We meticulously select the optimal step size $h$, for each model, consistently converging around a value around 1e-4.
>
> |         |         |  Recall |        |         |        | Precision |        |          |         |
> |---------|:-------:|:-------:|:------:|:-------:|:------:|:---------:|:------:|:--------:|:-------:|
> | Dataset |  Method | COV(%)↑ |        | MAT(Å)↓ |        |  COV(%)↑  |        |  MAT(Å)↓ |         |
> |         |         |   Mean  | Median |   Mean  | Median |    Mean   | Median |   Mean   |  Median |
> | Drugs   | GeoDiff |  42.35  |  33.30 |  1.3326 | 1.3339 |   21.62   | 13.17  | 204.4705 |  1.9867 |
> |         |  SDDiff |   1.24  |  00.00 |  6.6569 | 6.9824 |   00.44   |  00.00 | 291.7993 | 97.1702 |
> | QM9     | GeoDiff | 10.48   | 5.48   | 0.6390  | 0.6432 | 2.21      | 1.40   | 116.6899 | 90.4334 |
> |         | SDDiff  | 25.54   | 19.47  | 0.6018  | 0.6009 | 5.04      | 3.67   | 9.8198   | 3.6648  |

---

> ### Author Response · Authors · 2023-11-19
>
> **Questions**
> > 1. On  FrameDiff (Yim et al., 2023) in Section 2 and MAT-R  in Appendix G.
>
> Thanks for your pointing out it. We have revised it.
>
> > 2. On the top of page 5 , symbols α_ij and e_ij are introduced without explanation. Also, it would be nice to provide a brief explanation or reference to the generalized multidimensional scaling.
>
> These notations are commonly used in linear algebra.  $e_{ij}$ is a square matrix such that its entry in the i-th row and j-th column equals 1 and the other entries equal to 0. $\alpha_{ij}$ denotes the magnitude of change in edge that connects node $i$ and node $j$. The derivation for the Eq 8-9 can be found in the Appendix D.2.
> Generalized multidimensional scaling extends metric multidimensional scaling by accommodating a non-Euclidean target space. GMDS facilitates the discovery of the most accurate embedding of one surface into another when dissimilarities represent distances on a surface and the target space is an alternative surface.
>
> > 3. In the sentence just before eq 11 , the meaning of the phrase "the pairwise distance matrix is sparse" is not clear.
>
> This means that in practice, we usually consider a partially connected graph, hence, we only consider a proportion of entries in the full adjacent matrix.
>
> > 4. It might be better to reformulate the optimization problem in eq 8 by replacing the summation over all $i<j$ pairs with connected $(i, j)$ pairs.
>
> Thanks for your suggestions, we will consider modifying this.

---

### Official Review · Reviewer_bG3y · 2023-11-01

**Soundness:** 2 fair
**Presentation:** 2 fair
**Contribution:** 1 poor
**Rating:** 3
**Confidence:** 4

**Summary:**

In this paper, two modifications to the existing sampling methods of the diffusion model are proposed to improve the sampling rate for generating molecular conformations in SE(3) invariant spaces.

**Strengths:**

- This paper is well-organized.
- The paper focuses on accelerating the sampling process of the diffusion model for conformation generation, which is an interesting topic.

**Weaknesses:**

1. The paper is not well-motivated. The proposed sampling method is specifically designed for the task of molecular conformation generation in SE(3)-invariant space.  However, the SOTA method of conformation generation like Torsional Diffusion which operates on the hypertorus achieves much better performance than GeoDIFF which operates on Euclidean space. Moreover, the sampling process of Torsional Diffusion only needs 20 steps. Therefore, focusing on accelerating the sampling process of GeoDIFF/SDDiff seems meaningless.
2. The contribution of the introduced two minor modifications is limited. The operations just introduced two coefficients into existing sampling methods[1] to compensate for the poor performance on the conformation generation task by using [1] directly. From the experiments, the proposed methods don't get surprising results.
3. The paper is over-clams. The paper doesn’t provide the theoretical results about the statement “We analyze current modeling methods in SE(3) (Shi et al., 2021; Xu et al., 2022; Zhou et al., 2023) and theoretically pose crucial mistakes shared in these methods, which inevitably bring about the failure of acceleration” and don’t give the reasons why the modification solve the theoretical problems. If the proposed modifications can solve the problems, at least it can be directly used to improve the existing sampling methods applied in (Shi et al., 2021; Xu et al., 2022; Zhou et al., 2023).
4. The paper is not well-written. The paper hasn’t explained clearly why the introduced two modifications can accelerate the sampling process significantly. For modification one, the introduced $degree_i$ can make the sparse conformation into a fully connected conformation. Why connected conformation can outperform sparse one?
5. The proposed sampling method mainly improves the recall but gains worse performance in precision. Therefore, this method, which is only used for conformation generation, is not very practical.
6. The experimental results of baseline GeoDiff are significantly lower than the results reported in the GeoDiff paper.

[1] DPM-Solver: A Fast ODE Solver for Diffusion Probabilistic Model Sampling in Around 10 Steps

**Questions:**

See section weakness.

---

> ### Author Response · Authors · 2023-11-19
>
> > 1. The paper is not well-motivated. The proposed sampling method is specifically designed for the task of molecular conformation generation in SE(3)-invariant space. However, the SOTA method of conformation generation like Torsional Diffusion which operates on the hypertorus achieves much better performance than GeoDIFF which operates on Euclidean space. Moreover, the sampling process of Torsional Diffusion only needs 20 steps. Therefore, focusing on accelerating the sampling process of GeoDIFF/SDDiff seems meaningless.
>
> We do not agree with that accelerating the sampling process of GeoDIFF/SDDiff seems meaningless. Different from Torsional Diffusion, GeoDIFF and SDDiff are purely data-driven without any prior knowledge and preprocess(like RDkit in Torsinal Diffusion) and their distance-based modeling is effortless, indicating that they are not limited in the conformation generation tasks. As for our acceleration method, from the derivation of our method is also applicable to any coordinates generation tasks that take SE(3)-invariance into consideration. For example, our method is also applicable to sparse cloud point generation tasks. However, torsional diffusion is inapplicable in this case since it cannot first generate a local structure. In a nutshell, although focusing on molecular conformation generation, our contribution in deed lies in that we rethink the modeling occurring in the general SE(3)-based methods and propose a scheme for accelerating general diffusion-based generation in SE(3)-invariant space, and we plan to generalize it in future works.
>
> > 2. The contribution of the introduced two minor modifications is limited. The operations just introduced two coefficients into existing sampling methods[1] to compensate for the poor performance on the conformation generation task by using [1] directly. From the experiments, the proposed methods don't get surprising results.
>
> Directly using the DPM solver is inapplicable rather than just poor performance, we have demonstrated in Sec 4 around Eq.4 "...cannot produce substantial conformations. ". We found that the results are chaos. Ignoring any one of our proposed two modifications would lead to zero performance w.r.t COV and nan w.r.t. MAT.
> Our proposed modifications are minor but non-trivial. Such a modeling issue has existed since ConfGF (2021) and remains unsolved. The mathematical proofs behind such two simple modifications are intricate, and our experiments strongly proved that our method gains a roughly 50-150x speedup on comparable performance.
>
> > 3. The paper is over-clams. The paper doesn’t provide the theoretical results about the statement “We analyze current modeling methods in SE(3) (Shi et al., 2021; Xu et al., 2022; Zhou et al., 2023) and theoretically pose crucial mistakes shared in these methods, which inevitably bring about the failure of acceleration” and don’t give the reasons why the modification solve the theoretical problems. If the proposed modifications can solve the problems, at least it can be directly used to improve the existing sampling methods applied in (Shi et al., 2021; Xu et al., 2022; Zhou et al., 2023).
>
> The current modeling of the transformation from distance score to conformation score is heuristic. In Sec 4.1 we proposed revised modeling via differential geometry and proposed modification 1. Then we consider remedying the approximation error of the conformation score by an additional scaling factor in Sec 4.2.
>
> We do not understand what the reviewer means by ``at least it can be directly used to improve the existing sampling methods applied in (Shi et al., 2021; Xu et al., 2022; Zhou et al., 2023)''. We have directly used our method to accelerate the existing sampling methods applied in GeoDiff and SDDiff (Xu et al., 2022; Zhou et al., 2023) by two slight modifications, and the results are shown in experiments. In the experimental section, we showed that our method indeed improves the existing sampling methods since originally, using the traditional method to accelerate these models would all lead to 0 performance.
>
> > 4. The paper is not well-written. The paper hasn’t explained clearly why the introduced two modifications can accelerate the sampling process significantly. For modification one, the introduced $degree$ can make the sparse conformation into a fully connected conformation. Why connected conformation can outperform sparse one?
>
> Sec 4.1 and Sec 4.2 have introduced the theory behind our modifications. The first modification is not designed to transform the sparse conformation into a fully connected conformation. Instead, we argued that the previous computation of the conformation score is wrong, i.e., the conformation score cannot be directly computed by the chain rule. We proposed that when computing the conformation score, one of the sum operators should be replaced by a mean operator.

---

> > ### Author Response · Authors · 2023-11-19
> >
> > > 5. The proposed sampling method mainly improves the recall but gains worse performance in precision. Therefore, this method, which is only used for conformation generation, is not very practical.
> >
> > We think the decrease in precision metrics is acceptable considering that our method gains 50-100x speedup. Also, we have analyzed that the poorer performance in precision is due to the unstable network in Sec 5.4, which exceeds the boundaries of this investigation. And our model is not limited to conformation generation. Our proposed methods do not rely on any prior and given bias, thus it is a general method that can be used in any tasks that require SE(3)-invariant property.
> >
> > > 6. The experimental results of baseline GeoDiff are significantly lower than the results reported in the GeoDiff paper.
> >
> > First, it is noted that in our experiment we provided results of modified GeoDiff. We modified its modeling with the incorporation of modification 1, thus the results can be different.
> > Second, apart from modification 1, we have abandoned some tricks (maybe sacrifice the performance) in the original implementation of the GeoDiff to ensure the mathematical correctness of the diffusion process so that our acceleration method does not only apply to molecular generation tasks but also to other tasks that consider the SE(3)-invariance. For example, in the original implementation, two models are trained in the forward diffusion process, but only one part of the two models was used from the denoising time steps of $T$ to $T/2$ in the reversed diffusion process to generate conformers. Such a design does not mathematically fit the reversed diffusion formula. We revised these issues and retrained the model. Since we did not apply many tricks when training or sampling, the performance is weaker than that of the original model.
> > Finally, the main purpose of our experiments is to compare the performance of Langevin dynamics and our acceleration method, and the results are convincing enough to validate our method.

---

> ### Comment · Reviewer_bG3y · 2023-11-23
>
> Thanks for your reply and explanation. I'm maintaining my score because I don't feel that the paper's contribution is currently sufficient to be accepted by ICLR.

---

### Official Review · Reviewer_vTuX · 2023-11-05

**Soundness:** 2 fair
**Presentation:** 1 poor
**Contribution:** 2 fair
**Rating:** 3
**Confidence:** 4

**Summary:**

This work proposes novel strategies to improve diffusion-based molecular conformation generation models to achieve more accurate probabilistic modeling and faster generation. The proposed strategies include use average operation in coordinate score calculation, and multiplying a hyperparameter "scale" in generation. The proposed methods improves GeoDiff and SDDiff in some metrics of molecular conformation generation tasks.

**Strengths:**

Originality: The proposed strategies in this paper is novel.
Quality: Experiments on benchmark datasets show the porposed strategies can improve recall rates of GeoDiff and SDDiff in molecular conformation generation.
Clarity: No.
Significance: The proposed strategies can be useful and enlightening for developping score-based diffusion models for molecular conformation generation.

**Weaknesses:**

Major:
(1) Section 4.1 is not well-organized or logically smooth. It is hard to understand the overall process to obtain Equation (9) due to lack of clarification of many mathematical notations and motivations. Authors are encouraged to rewrite this section and give more clarification and explanation to the following questions:
- What are the meanings of TM, TN, $\pi_M, \pi_N$ in Figure 1?
- Why $f=\pi_N\circ d\phi = \phi \circ \pi_M$? What are the processes described by $\pi_N\circ d\phi$ and $\phi \circ \pi_M$?
- Why $\pi_M$ is chosen to be an identical mapping, $\pi_N$ is chosen to be a GMD?
- What does $e_{ij}$ mean in the formula of $\hat{d}$, and how the formula of $\pi_{M,\tilde{d}}(\hat{d})_{ij}$ and subsequently Equation (8) is derived?
- In Appendix D.2 (proof of theorem 1), what is $\lambda_{uv}$? Why does the second $\le$ (from line 30b-30c to line 30d-30g) hold?

(2) In Section  4.2, authors are recommended to clarify what is $\tilde{C}_{t+\lambda(s-t)}$?

Also, it would be better to discuss how much error will be introduced by the approximation $k_{s_\theta}(d_s, s, t) \approx k_{s_\theta}(p_{data})$?
(3) It will make the experimental results stronger if authors do ablation study to verify that every single strategy can improve the performance. Also, authors are recommended to make an in-depth discussion about why the proposed method achieves poor performance in precision metric.

Minor:
Typos: In abstract, "develop more precise approximate" --> "develop more precise approximation"
In Section 2 first paragraph, "so that the lengths of atom bounds" --> "such as the lengths of atom bonds"

**Questions:**

The proposed strategies are mainly used for diffusion models that calculate coordinate scores from distance scores. However, current state-of-the-art molecular conformation generation method, Torsional Diffusion [1], focusing on generating torsion angles. Can GeoDiff or SDDiff compete with Torsional Diffusion in generation speed when the proposed strategies are used? Can the proposed strategies or similar strategies be also applied to Torsional Diffusion?

[1] Torsional Diffusion for Molecular Conformer Generation. NeurIPS 2022.

---

> ### Author Response · Authors · 2023-11-19
>
> We thank reviewer vTuX for giving feedback timely. Reviewer vTuX mainly has concerns about notation definitions, while we have explicitly defined them in the Appendix. But possibly due to our writing, one may find these notations confusing. We will re-organize our formula in the final version so that notations become more straightforward. We present our response as follows.
>
> > 1. What are the meanings of $\mathrm{TM}, \mathrm{TN}, \pi_M, \pi_N$ in Figure 1?
>
> $TM$ and $TN$ are tangent space of the manifold of $M$ and $N$, respectively. $\pi_M$ and $\pi_N$ are the projection function that receives a vector in the tangent space and project it into the manifold space. A detailed introduction of the differential geometry definitions and notations can be found in Appendix C.
>
> > 2. Why $f=\pi_N \circ d \phi=\phi \circ \pi_M$? What processes are described by $\pi_N \circ d \phi$ and $\phi \circ \pi_M$?
>
> $f$ is defined to be the mapping from $TM$ to $N$. There are two ways to transform from $TM$ to $N$. One is by $TM \rightarrow TN \rightarrow N$ and the other one is by $TM \rightarrow M \rightarrow N$. The former transformation is defined by the composition $\pi_N \circ d \phi$ and the latter one is defined by $\phi \circ \pi_M$. Hence, $f=\pi_N \circ d \phi=\phi \circ \pi_M$. Such a $f$ function is only defined to compute $d \phi$ when the rest three functions are known. Such a method is commonly used in linear algebra such as when determining the properties of the quotient space.
>
> > 3. Why $\pi_M$ is chosen to be an identical mapping, $\pi_N$ is chosen to be a GMD?
>
> $C_t$ should always stay in the manifold $N$. Mathematically speaking, our reversed ODE should be replaced with a projected dynamical system in the form of $\frac{\mathrm{d} C_t}{\mathrm{d} t}=\Pi_N(C_t, -\frac{1}{2} \frac{\mathrm{d} \sigma_t^2}{\mathrm{d} t} \nabla_{C_t} \log q_{0 t}\left(C_t \right))$, where $\Pi_N(x, v)=\lim_{\delta \rightarrow 0^{+}} \frac{\pi_N(x+\delta v)-x}{\delta}$. Comparing with the exact update form $\frac{\partial C_t}{\partial t}=-\frac{1}{2} \frac{\mathrm{d} \sigma_t^2}{\mathrm{~d} t} \nabla_{C_t} \log q_{0 t}\left(C_t \mid C_0\right)$, we can take $\pi_N$ to be the identical mapping.
>
> $\pi_N$ is the projection from the tangent space to the manifold. By the definition of the projection, we must project a matrix to a feasible adjacent matrix while minimizing the distance under a certain metric, which is essentially the same as the problem of the GMD.
>
> > 4. What does $e_{i j}$ mean in the formula of $\hat{d}$, and how the formula of $\pi_{M, \tilde{d}}(\hat{d})_{i j}$ and subsequently Equation (8) is derived?
>
> $e_{ij}$ is a square matrix such that its entry in the i-th row and j-th column equals 1 and the other entries equal to 0. The derivation for the Eq 8-9 can be found in the Appendix D.2.
>
> > 5. In Appendix D.2 (proof of theorem 1), what is $\lambda_{u v}$ ? Why does the second $\leq$ (from line $30 \mathrm{~b}-30 \mathrm{c}$ to line $30 d - 30 g$ ) hold?
>
> $\boldsymbol{\lambda}_{u v}=\frac{\tilde{\mathbf{x}}_u-\tilde{\mathbf{x}}_v}{\left\|\tilde{\mathbf{x}}_u-\tilde{\mathbf{x}}_v\right\|}$, which can be found in the Appendix D.1. The second $\leq$ should be $=$. We apologize for this typo and have corrected this in the manuscript.
>
> > 6. In Section 4.2, authors are recommended to clarify what is $\tilde{C}_{t+\lambda(s-t)}$ ?
>
> After one update, we are expecting that the sample follows the distribution w.r.t $p_t$. However, since there is some inevitable error in the denoising process, we consider such noise acts like applying a forward diffusion process on $C_t$ so that the sample follows the distribution for some $p_{t+\lambda (s-t)}$, i.e., a time in the interval of $(t, s)$.
>
> > 7. Also, it would be better to discuss how much error will be introduced by the approximation $k_{s_\theta}\left(d_s, s, t\right) \approx k_{s_\theta}\left(p_{\text {data }}\right)$ ?
>
> This is a good question. Unfortunately, we do not know how to do this so far and we leave it as a future work.
>
> > 8. It will make the experimental results stronger if authors do ablation study to verify that every single strategy can improve the performance.
>
> Thanks for your suggestion, but it is not proper to conduct an ablation study. The main contribution of our work is to propose two **necessary** modifications for acceleration, which means that the absence of either one disables acceleration. Modification 1 (change ''add'' to ''mean'' on the transformation from distances to coordinates in Eq. 5) is a revision of the modeling to enable acceleration. In our experiment, the results are all from the models under Modification 1. Modification 2 (adds a multiplier ''scale'', $k_\mathbf{s}$ to the sampling process) is necessary for fast sampling. We found that without $k_{\mathbf{s}_{\boldsymbol{\theta}}}$, the fast sampling, i.e. sampling via DPM-Solver's proposed iteration in Eq. 4 will fail, and we conduct sampling via Langevin Dynamics (LD).

---

> ### Author Response · Authors · 2023-11-19
>
> > 9. Also, authors are recommended to make an in-depth discussion about why the proposed method achieves poor performance in precision metric.
>
> We have put the analysis on Precision metrics on Sec.5.4. ''This is due to the deteriorating output of the network.'' We observe that part of the output occasionally starts exploding from a certain sampling iteration, which leads to some unstructured conformers corresponding to one molecule. The recall metrics are not affected much because they only consider the coverage status of ground truth, while precision metrics calculate all generated conformers. This phenomenon is observed for both LD sampling and our fast sampling method, especially when the step size or scale $k_{\mathbf{s}_{\boldsymbol{\theta}}}$ is higher. More low-quality conformers are generated while sampling via the acceleration method, it is reasonable because fast sampling requires a larger step in each iteration, thus the precision metrics are worse. We think a stronger network or special design can solve the deteriorating output, but this is beyond the scope of this study. We'd like to conduct more stronger networks in further work.
>
> > 10. The proposed strategies are mainly used for diffusion models that calculate coordinate scores from distance scores. However, current state-of-the-art molecular conformation generation method, Torsional Diffusion [1], focusing on generating torsion angles. Can GeoDiff or SDDiff compete with Torsional Diffusion in generation speed when the proposed strategies are used? Can the proposed strategies or similar strategies be also applied to Torsional Diffusion?
>
> Current SOTA Torsional diffusion uses the RDkit to first generate the local structures and then apply the diffusion process on the torsional angles. Since they use the RDkit to generate the local structures, the main difficulties: considering the SE(3)-invariance is avoided. However, our method is a purely data-driven method, which cannot easily avoid the SE(3)-invariance. Hence, the computational speed is relatively slower than Torisonal diffusion's. Since the modeling manifold of the torsional diffusion is the torsion angle, which is greatly different from the distance manifold, hence, our method is inapplicable. However, for those generation tasks like cloud point generation, which takes SE(3)-invariance into consideration, our method applies to accelerate the reversed diffusion process.
>
> > 11. About typos.
>
> Thank you for bringing those typos to our attention. We are committed to making corrections.

---

> > ### Comment · Reviewer_vTuX · 2023-11-19
> > **Follow-up Response**
> >
> > I appreciate authors' work in addressing my questions and concerns.
> >
> > - As agreed by authors, the math notation and organization of methodology part, particularly Section 4.1, is not good and unfriendly to readers who are not familiar with differentiable geometry. Hence, I strongly encourage authors to rewrite this part and make them organize well. I do not think leaving the definitions of some notations to Appendix is a good idea. I encourage authors to upload the revised version of the paper if it is ready.
> > - By asking for an in-depth discussion about why the proposed method achieves poor performance in precision metric, I am actually asking reasons why "part of the output occasionally starts exploding from a certain sampling iteration". Particularly, this phenomenon happens for both the proposed method and LD, but it seems to happen more frequently for the proposed method, as the precision metric is lower than LD baseline. It leads to some concerns that the the proposed method accelerates the generation speed with a cost of significantly decreasing the model robustness and the quality of generated conformations. I hope authors can give an in-depth discussion to it, and some theoretic analysis about model robustness (the frequency of output exploding) if possible.

---

> > > ### Author Response · Authors · 2023-11-21
> > >
> > > > About the organization of the methodology part
> > >
> > > Thank you for bringing us to notice that too many unfamiliar definitions and notations only appear in the Appendix. In our final version, we will reorganize the methodology part and make it more straightforward.
> > >
> > > > Why does a part of the output occasionally start exploding from a certain sampling iteration?
> > >
> > > In the original implementation of the GeoDiff and SDDiff, **authors used some tricks to avoid exploding but after applying these tricks, the reverse diffusion modeling is not mathematical correct anymore. We did not apply these tricks concerning such acceleration may be applicable to a border domain**. In the original implementation of the GeoDiff and SDDiff, these models both consist of two parts, the global part $s_1$ and the local part $s_2$. During the training process, both the global and local parts are used for prediction, that is $s_1 + s_2 = \epsilon$. However, at the beginning of the sampling process, if both parts are used, the model occasionally explodes. Such a phenomenon is also observed by GeoDiff's authors. Hence, they only use local part $s_2$ at the beginning of the sampling process, i.e. $s_2 = \epsilon$ until a certain timestamp, they add the global part $s_1$ back, which does not fit the mathematical formulation of the diffusion. To make the model applicable to a border domain, we correctify such a misfit but make the model occasionally explode.

---

### Meta-Review · Area_Chair_enGH · 2023-12-04

**Metareview:**

The paper tries to improve and accelerate diffusion-based conformation generation for molecules. Methodologically the paper derived the formulation of score model on the distance space, analyzed approximation errors, and made modifications to the generation process for acceleration. Empirically the paper showed significant acceleration for generation. Critical weaknesses include unclear and seemingly sloppy mathematical formulation of the problem, noticeable deficiency in the precision score of the generation results, and limited potential practical interest in the domain compared to existing methods (e.g., Torsional Diffusion).

**Justification For Why Not Higher Score:**

All reviewers found that the mentioned weaknesses make the contribution less solid and significant, and lead to concerns themselves. Authors rebuttal did not effectively address the critical concerns.

**Justification For Why Not Lower Score:**

N/A

---

### Decision · Program_Chairs · 2024-01-16

Reject